# Spatially selective delivery of living magnetic microrobots through torque-focusing

Nima Mirkhani [1], Michael G. Christiansen [1], Tinotenda Gwisai [1], Stefano Menghini[1] & Simone Schuerle [1] ✉

Rotating magnetic fields enable biomedical microrobots to overcome physiological barriers and promote extravasation and accumulation in tumors. Nevertheless, targeting deeply situated tumors requires suppression of off-target actuation in healthy tissue. Here, we investigate a control strategy for applying spatially selective torque density to microrobots by combining rotating fields with magnetostatic selection fields. Taking magnetotactic bacteria as diffuse torque-based actuators, we numerically model off-target torque suppression, indicating the feasibility of centimeter to millimeter resolution for human applications. We study focal torque application in vitro, observing off-target suppression of actuation-dependent effects such as colonization of bacteria in tumor spheroids. We then design and construct a mouse-scale torque-focusing apparatus capable of maneuvering the focal point. Applying this system to a mouse tumor model increased accumulation of intravenously injected bacteria within tumors receiving focused actuation compared to non-actuated or globally actuated groups. This control scheme combines the advantages of torque-based actuation with spatial targeting.

Delivering therapeutic payloads or immune-modulating living vectors to sites where they are needed remains a perennial challenge, especially in cancer therapy[1]. The efficiency of delivery as a fraction of administered dose continues to be limited, even for carefully designed nanoparticles (NPs) that act as carriers employing physiochemical targeting strategies[2–5]. Similar hurdles have been encountered by self-propelled therapeutic bacteria, for which delivery efficiency is especially crucial because tolerable doses are low[6–9]. Recently, the scope of investigation into biomedical microrobots suited for drug delivery has expanded remarkably[10–12]. Microrobots designed to convert externally applied magnetic stimuli to locomotion, fluid convection, or other modes of therapeutic activity are particularly useful for deep physiological targets, which magnetic fields can access unencumbered[13–16]. Historically, magnetic field gradients that apply forces to magnetic materials introduced into the body have been envisioned for targeting[17,18], an approach that is fundamentally limited to superficial targets in the absence of additional constraints[19]. More recently, uniform fields that steer self-propelling microrobots[11,20–23] or rotating

fields that power motion through applied magnetic torques[24–29] have offered compelling alternatives that are suited for deep tissue targeting at the human scale (Fig. 1A). We previously showed how rotating magnetic fields (RMFs) can be applied to naturally magnetic bacteria, known as magnetotactic bacteria (MTB), to help them overcome physiological barriers to reach tumors. MTB can thus be seen both as biological microrobots and potential vectors in bacterial cancer therapy when their magnetic properties are leveraged in a hybrid control strategy[30].

Given that RMFs, when applied to diffuse dispersions of synthetic or living microrobots, are capable of enhancing extravasation and tissue penetration, it follows that a strategy to locally focus magnetic torque density is needed to reduce off-target actuation. Ideally, this would utilize a magnetic stimulus that does not require detailed knowledge of the instantaneous distribution of microrobots, but rather serves as a form of open-loop control, selectively transferring torque density to microrobots within a target region as they circulate throughout the body (Fig. 1A). To accomplish this, a magnetostatic

[1]Institute for Translational Medicine, Department of Health Sciences and Technology, ETH Zurich, CH-8092 Zurich, Switzerland.
✉e-mail: simone.schuerle@hest.ethz.ch

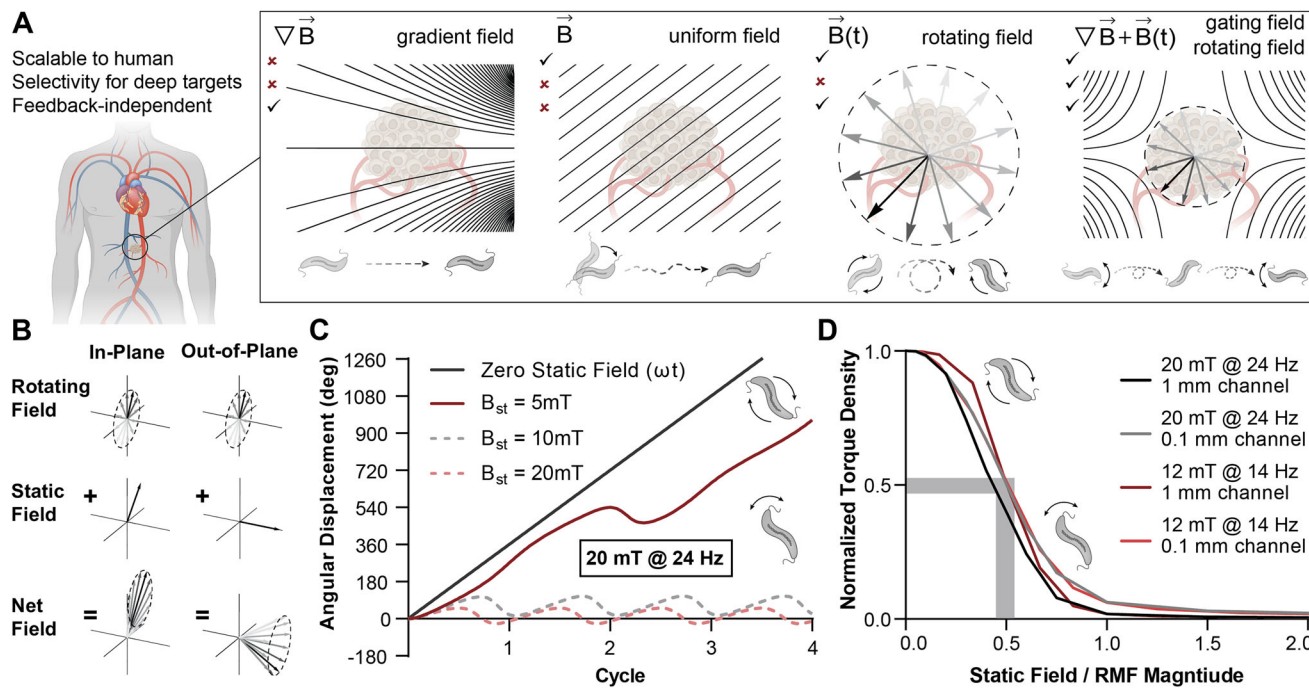

**Fig. 1 | Concept of magnetic torque-focusing by combining RMFs and magnetostatic selection fields. A** Overview comparing the advantages and disadvantages of various forms of magnetic field application to control vascularly dispersed magnetic microrobots, here depicted as MTB, our model microrobot. Possible magnetic stimuli include gradient forces, uniform fields for steering, global RMFs, and RMFs with selection fields. **B** Vector superposition of magnetostatic fields and RMFs with a relative magnitude of 1.5 times for the special cases of a magnetostatic field vector in the plane of rotation (in-plane) and perpendicular to the plane of rotation (out-of-plane). **C** Angular displacement of simulated individual MTB

microbots exposed to different magnitudes of the static selection field $B_{st}$ with RMF of 20 mT and 24 Hz. **D** Simulated nondimensionalized torque density curves of bacteria under different geometric and actuation conditions. The curves nearly overlap, suggesting a generalized interpretation of off-target torque density suppression as determined by the relative magnitude of the magnetostatic field and RMF. Source data are provided as a Source Data file. Graphical elements depicting bacteria have been adapted from ref. 46. Graphical elements depicting tumors have been adapted from ref. 30. Reprinted with permission from AAAS.

selection field that supplies a field-free point or field-free line can be introduced, which suppresses torque-based actuation outside these regions[30–34]. This concept is adapted from magnetic particle imaging (MPI), where acquisition of inductive signals from diffuse magnetic nanomaterial is locally confined to image voxels by a similar selection field[35–37]. Research has indicated the possibility of using magnetostatic fields superimposed with high frequency (100 s of kHz) alternating magnetic fields to spatially restrict hysteretic heat dissipation of magnetic nanomaterials[33,38–42]. More recently, the feasibility of integrating RMF with static selection fields has also been shown[31,43–45]. In a pioneering work, spatially selective control of centimeter-scale objects such as screws has been achieved using MPI instrumentation[31]. Similarly, spatial control of rolling magnetic NP swarms within a tube was investigated[44]. While these studies demonstrated the compatibility of RMF with selection fields for targeted magnetic actuation, our strategy does not rely on feedback and is compatible with diffuse agents at low concentrations, and thus, is well suited for systemic drug delivery.

Here, we make use of selection fields as an integral component of a spatially targeted open-loop control strategy for diffuse living microrobots via torque-based actuation. In this method, the spatial focusing of torque density reduces the problem of targeted actuation to adjusting the size and position of the field-free point with respect to the targeted tissue, eliminating the need to individually track microrobots. Using an in vitro platform based on a commercial multi-axis electromagnet setup in combination with an array of small permanent magnets, we demonstrate the possibility for localized actuation of MTB within a targeted region. We show how this targeted actuation influences translational velocity of the MTB, convection-enhanced penetration of non-magnetic NPs into collagen matrices, and RMF-enhanced colonization of tumor spheroids. Encouraged by these

results, we designed and constructed a torque-focusing setup for small rodents, consisting of an external array of ferrite and neodymium iron boron (NdFeB) permanent magnets arranged as modified "magic sphere", an internal three-phase RMF generator, and a pair of DC coils to controllably move the field-free point. The position of the field-free point along the longitudinal axis of the animal is maneuvered by a positioning track, while its position within the transverse plane is adjusted electronically by currents within the DC coils. This device is validated through characterization of the net fields experienced in the working volume and through a magnetic mixing experiment that verified spatially selective fluidic convection. Finally, a small in vivo study was performed to compare the accumulation of MTB in flank tumors in mice exposed to global RMF actuation, spatially selective RMF actuation, and non-actuated conditions. Our approach represents a further advantage of RMF-based actuation of biomedical magnetic microrobots and provides a means to increase spatial selectivity in the effort to address longstanding challenges in targeted drug delivery.

## Results

### Combining rotating and magnetostatic selection fields focuses torque application

Magnetic microrobots introduced intravenously typically experience rapid dispersion throughout the body by the circulatory system, and magnetic stimuli can then be applied as part of a control strategy to influence their distribution. Although many approaches for magnetic control are possible, an ideal control scheme should not require knowledge of the instantaneous distribution of the microrobots, should scale well to form factors compatible with human patients, and should be able to specifically target remote points. Combining RMFs with magnetostatic selection fields for selective torque-based

actuation[30,31,33] represents a way to simultaneously satisfy these requirements when applied to diffuse distributions of microrobots (Fig. 1A). The principle underpinning the spatial selectivity is that a superposition of an RMF and a magnetostatic field leads to the suppression of the rotational character of the net field (Fig. 1B). The two representative examples shown in Fig. 1B are for the special case of an in-plane magnetostatic contribution, which shifts the circle of rotation, and for the special case of an out-of-plane magnetostatic contribution, which produces precession. At arbitrary points within a typical selection field, behavior is usually intermediate between these cases.

To elucidate the transfer of torque from a superimposed RMF and selection field to microrobots, the influence of the field can first be considered numerically on the level of individual MTB. These naturally magnetic bacteria contain a stably magnetized chain of magnetite nanocrystals surrounded by a soft bacterial body that experiences viscous drag from the surrounding fluid, which we have previously modeled[30]. Figure 1C shows the angular displacement of an idealized MTB over time under increasing in-plane magnetostatic contributions in a 20 mT RMF, with a similar simulation for a 12 mT RMF shown in Fig. S1A. If the magnitude of the magnetostatic component is small compared to the magnitude of the RMF, the bacterium is predicted to undergo a full rotation, although the angular velocity can vary in time (Fig. 1C and Fig. S1A). As the magnitude of the selection field increases, full rotation is suppressed, and the angular displacement oscillates with increasingly limited amplitude during the cycle of the RMF.

Whereas a constant magnetic torque is expected for an MTB rotating in steady state in response to an RMF in the absence of any selection field, introducing a magnetostatic component results in time-dependent torque that changes sign throughout the cycle of the RMF but remains nonzero when averaged over the cycle (Fig. S1B). As the magnitude of the magnetostatic component increases and full rotation is suppressed, the time-averaged torque approaches zero.

Because the results obtained for models of isolated MTB do not account for their interactions with each other and with nearby boundaries, the influence of magnetostatic selection fields on time-averaged torque density transferred to MTB suspensions confined to microchannels was considered with a previously established continuum model[32]. In the model, torque density was applied under various imposed magnetostatic field magnitudes and directions. Channel geometries of different scale were examined (100 μm and 1 mm), along with two distinct step-out conditions (20 mT at 24 Hz and 12 mT at 14 Hz), as shown in Fig. S1C, D. When normalized to their maximum value, averaged over all imposed magnetostatic field directions, and plotted in terms of the relative magnitude of the magnetostatic field and RMF, the time-averaged normalized torque density falls onto a similar characteristic curve (Fig. 1D). This curve reveals the basis for spatial selection in this system, exhibiting a nonlinear drop-off of the transferred torque density as a function of the relative magnitude of the magnetostatic selection field with the RMF. Approximately 50% of the torque density is predicted to be suppressed when the magnetostatic component reaches half the magnitude of the RMF.

### Selective actuation of MTB locally targets accumulation of co-suspended NPs

As a first step toward empirically observing the influence of combined magnetostatic selection fields and RMFs on the actuation of MTB, we performed a set of small-scale proof-of-principle experiments with our microscope-compatible arbitrary magnetic field generator. Because we have previously shown that convection generated by MTB subjected to RMFs can observably increase the transport of co-suspended NPs into collagen matrices[46], we hypothesized that this effect could be spatially restricted through the superposition of a magnetostatic selection field. Additionally, since the NPs were non-magnetic, their enhanced transport is predominantly governed by torque-induced fluid convection, highlighting the impact of torque-focusing compared to any gradient-based mechanisms. To test our hypothesis, a microfluidic chip was designed with five 4-mm-diameter circular chambers arranged in a cross (Fig. 2A). Circular contact lines[47] were incorporated into the center of each chamber (Fig. S2A), enabling the confinement of collagen gels in the center and formation of a barrier-free interface with the surrounding liquid compartment. Successful compartmentalization within the chambers was demonstrated by TAMRA-labeled collagen and suspension of green fluorescent NPs (Fig. 2A). As a source for the selection field, four small rectangular NdFeB magnets with opposing magnetization directions were stably held on the sides of the device by holes cut into the PDMS (Fig. 2B). This arrangement produced a field-free region coinciding with the central chamber of the device. By varying the geometry of the permanent magnets, selection fields with different gradients (4 T/m and 8 T/m), and thus different spatial targeting resolutions were produced (Fig. 2B and Fig. S2B). As an initial test of spatially restricted torque application, the translational velocities of MTB in the absence of collagen in the various chambers of the 4 T/m device were observed in response to an RMF at 12 mT and 14 Hz. Velocity in the targeted central chamber was compared to the surrounding off-target chambers, indicating significant suppression of transferred torque density away from the field-free region (Fig. 2C). For subsequent experiments, the device with the lower magnetostatic field gradients was selected because the resolution of its target point was sufficient for the size of the central chamber and the smaller magnets reduced interactions with the multi-axis electromagnets responsible for RMF.

Before experimentally examining the influence of focused torque transfer to MTB on co-suspended NP transport, concentration profiles within the liquid compartments were modeled numerically. The results, shown in Fig. 2D and Fig. S3, indicate that one important consideration in this experiment is the choice to either keep the plane of rotation of the RMF fixed in a single out-of-plane direction relative to the device or to sweep the plane of rotation continuously through the full circle of all possible out-of-plane rotations. A fixed plane of rotation is predicted to lead to nonuniform concentration profiles in the liquid compartment, since sustained convection in a particular direction breaks the symmetry of the device (Fig. 2D). This effect should be mitigated by continuously varying the direction of rotation, and more uniform enhanced convection throughout the device is also expected to modestly increase overall NP accumulation in the collagen (Fig. 2E).

Experimentally, both a sweeping RMF and fixed RMF were employed so that their influence on NP diffusion into the central collagen matrix could be compared. For the transport experiments, in contrast to Fig. 2A, unlabeled collagen was used in the center of the chambers and MTB suspension containing 200 nm red fluorescent non-magnetic NPs was placed in the surrounding compartment, so that fluorescence intensity could be used to infer concentration before and after the actuation with minimal crosstalk due to green background autofluorescence (Fig. 2F and Fig. S4). Quantification of NPs present inside the collagen area showed a significant increase in transport under magnetic actuation for both sweeping and constant RMF in the absence of a selection field compared to a negative control relying only on passive diffusion (Fig. 2G). When a selection field was applied, NP transport into the off-target chambers was reduced to a level comparable to passive diffusion, whereas more than 40% of the magnetically enhanced transport was maintained within the target region. One possible source of the reduction in transport at the target point is the role of gradient forces within the experimental setup. Whereas the magnetostatic field applied here is intended primarily to extinguish torque density, its associated gradients can also apply forces to MTB that draw them to the boundary of the chambers and interfere with rotational actuation. The respective average magnitudes of the relevant forces can be estimated to be 0.5–1.0 pN for torque-

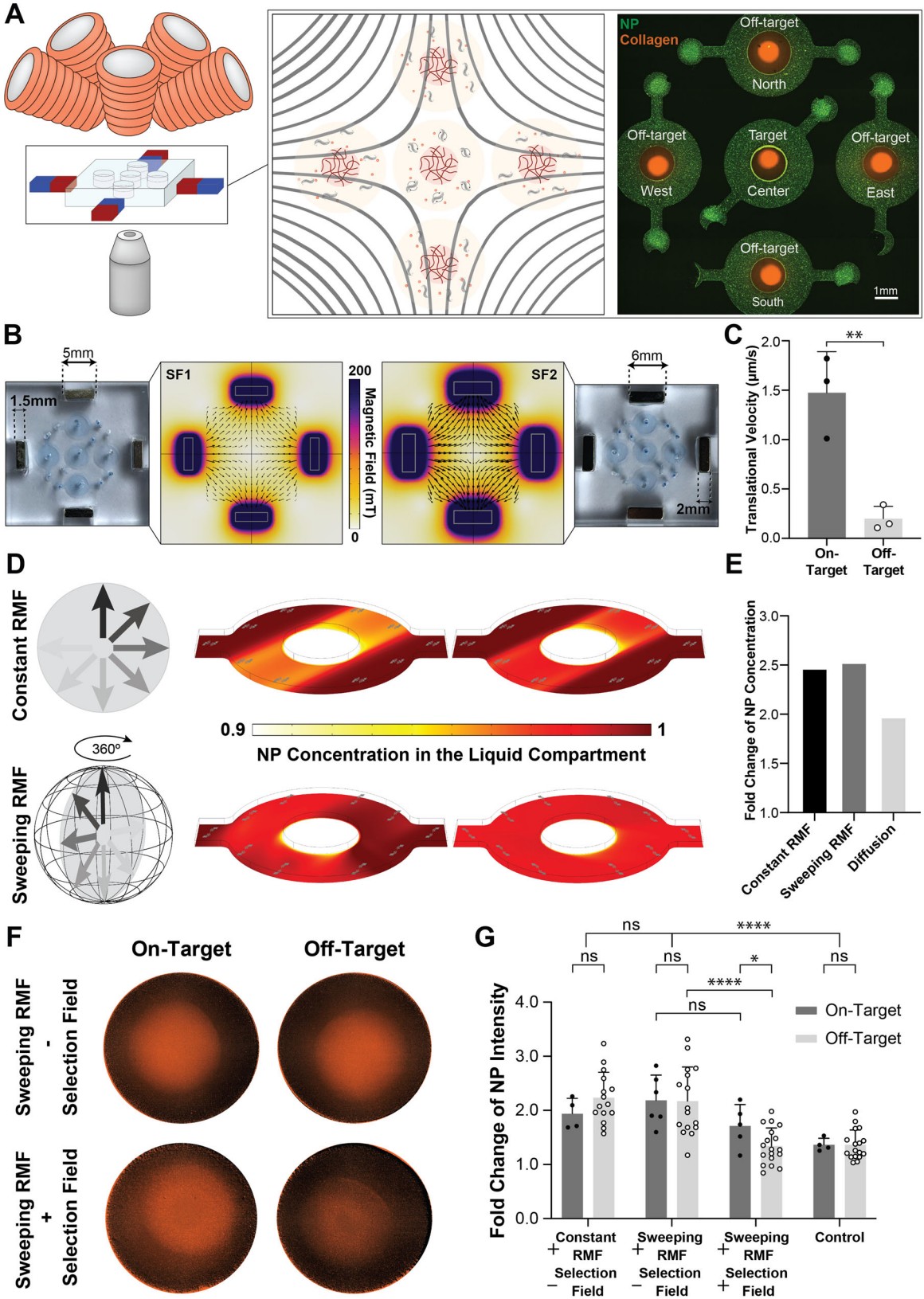

based forces[30], 0.1–0.5 pN for intrinsic bacterial propulsive forces[30], and 0.005 pN for magnetic gradient forces (given an upper bound of 10 T/m). Since the gradients in this miniaturized proof-of-concept setup are approximately twice that of the highest gradient realized in the larger setup described in a later section, these forces, while comparatively small, are expected to play a more dominant role here.

## Focused torque application selectively increases tumor spheroid colonization of MTB

As a physiologically relevant model of solid tumors, the miniaturized setup described in the previous section was adapted to study whether a magnetostatic selection field could spatially control bacterial colonization in tumor spheroids (Fig. 3A). We have previously found

**Fig. 2 | Selective actuation of MTB and localized convective effects in micro-fluidic devices. A** Schematic of the experimental setup composed of an electro-magnetic system and a microfluidic device containing five chambers, around which four permanent magnets are fixed to produce a field-free point coinciding with the target chamber. The chambers offer a circular tissue-fluid interface enabled through a contact line pinning method confining collagen (red) surrounded by suspensions of NPs (green). **B** Computational modeling of the selection fields (SF1 and SF2) generated by two differently sized NdFeB magnets. The larger set of magnets generates a stronger field and thus achieves a higher spatial resolution, but leads to higher gradients (8 T/m versus 4 T/m). **C** Translational velocity of MTB in off-target chambers vs the target chamber, demonstrating off-target suppression ($n = 3$ biological replicates; means ± SD; *$p < 0.05$, **$p < 0.01$, independent samples $t$ test, two-tailed). **D** Contours of NP concentration in the liquid compartment under constant and sweeping RMF. Constant RMF leads to the emergence of a separate band inside the fluid while sweeping RMF vectors lead to a uniform increase in the transport through an effectively larger surface area. **E** Simulated fold change of NP concentration in the collagen compartment for unactuated controls vs samples exposed to sweeping and constant RMF, respectively. **F** Representative images of NP distributions within the collagen compartment in target and off-target chambers, under sweeping RMF with and without the application of a selection field. **G** Fold change of NP fluorescent intensity in the collagen compartment for NP-MTB suspensions under torque-based actuation with and without a selection field and sweeping vs constant RMFs compared to unactuated, diffusion-based controls (independent experiments; on-target: $n = 4$ Constant RMF−SF, $n = 6$ Sweeping RMF −SF, $n = 5$ Sweeping RMF + SF, $n = 4$ Control; off-target: $n = 15$ Constant RMF−SF, $n = 16$ Sweeping RMF−SF, $n = 18$ Sweeping RMF + SF, $n = 17$ Control; means ± SD; *$p < 0.05$, **$p < 0.01$, ***$p < 0.001$, ****$p < 0.0001$, independent samples $t$ test, two-tailed). Application of the selection field suppresses off-target NP transport while only a nonsignificant reduction in the target chambers is observed. Source data are provided as a Source Data file. Graphical elements (electromagnet) have been adapted from ref. 46. Reprinted with permission from AAAS.

that torque-based actuation of MTB increases their colonization of tumor spheroids[30], a widely used in vitro 3D tumor model[48,49]. Each spheroid is composed of a mass of densely packed cancer cells that mimics some of the most relevant features of actual tumors, including cell-cell interactions and oxygen and nutrient gradients. Here, human breast cancer cells, MCF-7, were used to form spheroids roughly 400 μm in diameter, which were placed in 3 mm holes punched in PDMS in a well pattern mimicking the previous device (Fig. 3B). Spheroids were added to the wells with 25 μl of bacterial suspension of fluorescently labeled MTB for 1 h of magnetic actuation followed by collection, washing, and incubation for 1 day. This combination can be seen as a hybrid control strategy where magnetically enhanced delivery precedes taxis-driven penetration. Bacteria penetrating a spheroid are expected to encounter additional resistance from interactions with solid components of the spheroid, and we have found this effect to be comparable to an increased effective viscous drag, both experimentally (Fig. S5) and through simulations (Fig. S6). Accordingly, the RMF conditions selected to actuate the MTB here were 20 mT at 14 Hz. To reduce the magnetic gradient forces experienced by the MTB, a consideration explained in the previous section, the device featuring smaller magnets (4 T/m gradient) was again used for these experiments. However, an increased RMF magnitude relative to the magnetostatic field translates to a slightly lower resolution, i.e., a larger field-free region. This change is expected to shift the balance in the trade-off that was observed between off-target suppression and on-target transport.

After actuation and incubation, confocal fluorescence microscopy was used to analyze the accumulation of MTB within the spheroid, with Z-stacks taken at 10 μm intervals up to a total height of 200 μm (Fig. 3C). The MTB was labeled with a far-red pro-liferative dye to ensure that daughter cells resulting from pro-liferation during incubation could also be detected. In the absence of a selection field, significantly higher accumulation (2.3 fold) was observed after exposure to sweeping RMFs compared to the unactuated control (Fig. 3D). With the introduction of a selection field, MTB colonization of the on-target spheroids did not drop significantly, whereas a significantly suppressed accumulation was observed at off-target sites in the presence of the superimposed selection field (Fig. 3D). Clusters of bacteria inside the tumor spheroids were visible after 24 h of incubation, as shown in the representative Z-projected confocal images of the tumor spheroids depicted in Fig. 3E. Lastly, it is worth highlighting that the altered resolution within the same setup allowed a similar degree of colonization at the target site to be maintained with or without the selection field. For a miniaturized setup like the one employed here, this comes at the cost of incomplete suppression of off-target effects.

## Engineered mouse-scale setup demonstrates scalability of focused torque application

The in vitro proof-of-concept experiments described in the previous two sections demonstrated the feasibility of combining selection fields with RMFs for spatially restricted torque-based actuation. To take steps toward testing these concepts in vivo and exploring potential form factors for scaled up instrumentation, a mouse-scale setup was designed and constructed. The specific needs of the foreseen in vivo experiments imposed several key design requirements. Firstly, the working volume of the RMF should encompass the entire animal and its magnitude should be in the range of 10–20 mT for comparability to the in vitro experiments. Next, the volume of the desired field-free region formed by the selection field should be ~1 cm³ to coincide appropriately with a flank tumor. Lastly, the ability to manipulate the position of the field-free region with respect to the mouse was also desired to ensure that it could be accurately targeted to the tumor.

An exploded view of the design that was developed to meet these requirements is shown in Fig. 4A, along with its assembled form including a representation of an anesthetized mouse in Fig. 4B. It consists of three main components: 1) an array of permanent magnets that produces a field-free point (Fig. S7), 2) a set of AC coils that are used to generate the RMF (Fig. S8), and 3) a DC offset coil that moves the zero point within the transverse plane of the animal and rejects heat through forced water cooling (Fig. S9).

The array of permanent magnets responsible for generating a magnetostatic selection field was based on a "magic sphere" geometry, formed by an axisymmetric revolution of a second-order Halbach cylinder[50] (Fig. 4C). The desired distribution of magnetization was approximated with 2 cm × 2 cm × 2 cm stacks consisting pre-dominantly of ferrite ceramic permanent magnets (grade Y35) affixed to a 3D printed frame that held them at the appropriate angles. To ensure access to the working volume, magnets were omitted from the poles of the magic sphere, and sintered NdFeB magnets (N45) were strategically incorporated around the equator of the array to improve the isotropy of the field-free point. The results of an axisymmetric finite element model show the expected distribution of magnetostatic field magnitude, with a central field-free region that is an oblate spheroid with its shortest axis coinciding with the axis of symmetry.

Strong RMFs are routinely generated at the scale of tens of cen-timeters for industrial applications such as 3-phase industrial motors, which acted as a source of inspiration for the RMF coil geometry in this setup. Three independent stators were wound around a 3D printed frame and driven with 3-phase current to produce an adequately homogenous RMF, with its plane of rotation perpendicular to the axis of the cylindrical working volume (Fig. 4D). The target field magnitude of 20 mT was achieved in the 5.5 cm diameter, 8.5 cm long working volume, but continuous operation was limited in practice to 15 mT based on heat rejection capabilities.

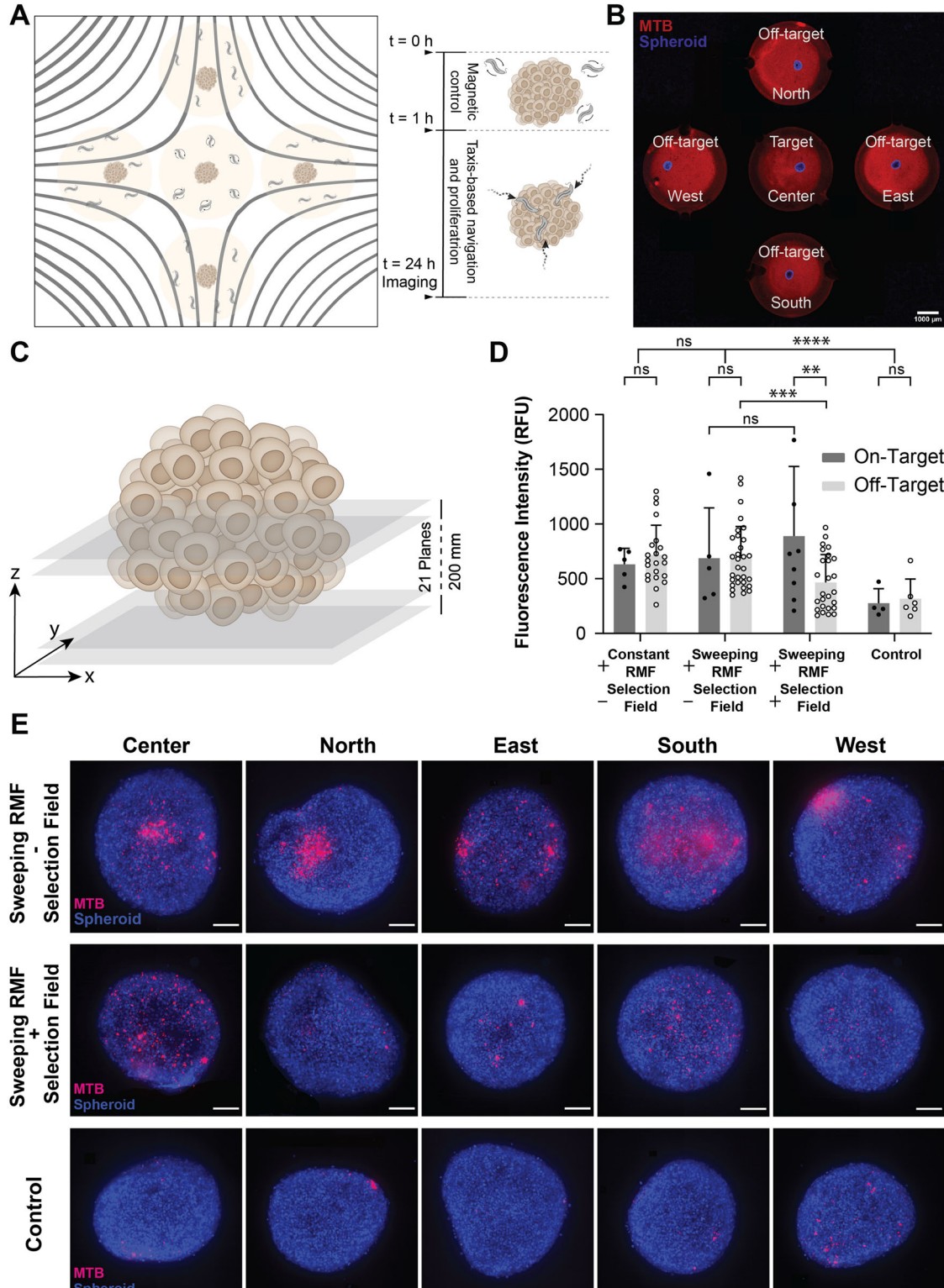

**Fig. 3 | Focused torque application enhances tumor colonization of MTB in target locations in vitro. A** Schematic of the experimental setup using the same configuration as above, but five open wells were provided instead of closed chambers to allow for convenient placement of 3D tumor spheroids cultured from MCF-7 cells. **B** Spheroids (blue) surrounded by a dispersion of MTB (red) labeled with a NIR proliferation dye allowing to track daughter cells upon proliferation. **C** Schematic of the readout procedure integrating the fluorescent intensity of Z-stacks at intervals of 10 μm for a distance of 200 μm along the z-axis. **D** Results of the analysis showing suppressed colonization in off-target spheroids compared to target spheroids (biological replicates; on-target: *n* = 5 Constant RMF−SF, *n* = 5

Sweeping RMF−SF, *n* = 9 Sweeping RMF + SF, *n* = 4 Control; off-target: *n* = 22 Constant RMF−SF, *n* = 31 Sweeping RMF−SF, *n* = 26 Sweeping RMF + SF, *n* = 6 Control; means ± SD; *\*p* < 0.05, *\*\*p* < 0.01, *\*\*\*p* < 0.001, *\*\*\*\*p* < 0.0001, independent samples *t* test, two-tailed). Sweeping RMF consistently yields a higher colonization than constant RMF. **E** Representative fluorescent micrographs of projected z-stacks for each experimental condition (scale bars: 100 μm). Source data are provided as a Source Data file. Graphical elements depicting bacteria have been adapted from ref. 46. Graphical elements depicting tumors have been adapted from ref. 30. Reprinted with permission from AAAS.

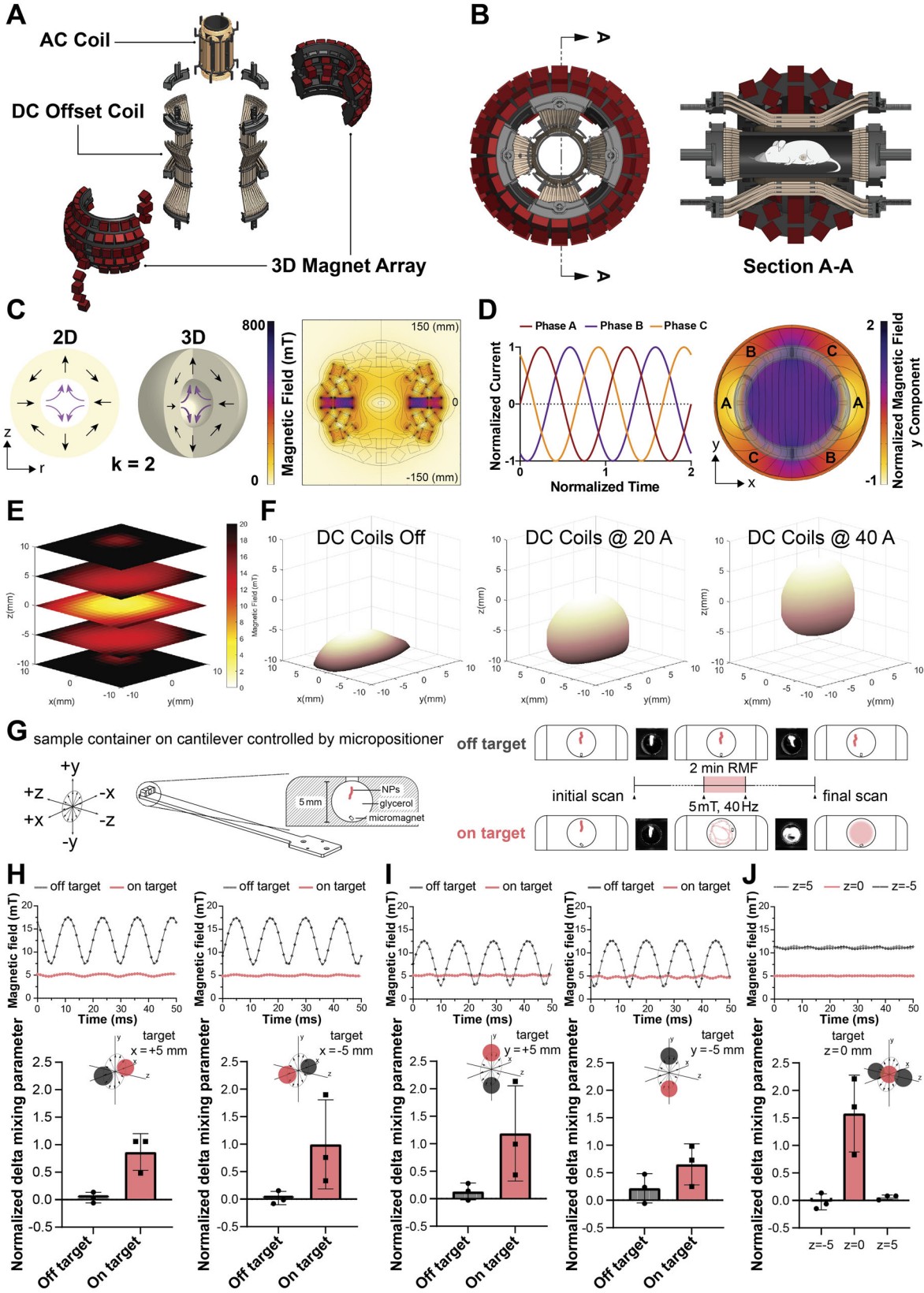

The two DC offset coils were arranged in the same basic concentric geometry as the RMF coils, residing between the RMF coils and the permanent magnet array (Fig. 4A). These coils were formed manually from 3 mm copper tubing and arranged geometrically using a set of 3D printed guides and flow splitting manifolds (Fig. S9). Electrical current was run in series through these coils, whereas chilled water flowed in parallel to cool the setup to temperatures safe for anesthetized rodents. Thermal contact between the RMF coils and the DC offset coils was made via electrically non-conductive thermal epoxy.

The obtained field-free point was spatially characterized with a fixed array of 64 individual Hall probes, with its ellipsoidal shape found

**Fig. 4 | Characterization and validation of a mouse-scale torque-focusing apparatus. A** Exploded 3D view of the device, showing the three main components labeled. **B** Assembled 3D view of the device, shown in cross section with a representation of a mouse. Inner diameter of the working volume is 5.5 cm. **C** The concept of a 3D "magic sphere" geometry is derived from an axisymmetric revolution of a second order ($k$ = 2) Halbach cylinder (left). Computational modeling indicates the expected magnitude of the field produced by the modified geometry actually used (right), which excluded magnets from the poles to preserve access and used a combination of ferrite ceramic and NdFeB magnets to improve isotropy of the zero point. **D** 120° phase shifted input currents are used to drive the three-phase RMF coil and generate a sufficiently uniform RMF. Computational modeling of the field generated by three phase stators around a transverse section of the RMF coil shows the expected uniformity of the field. **E** An array of 64 Hall probes was used to measure and reconstruct the field-free region of the permanent magnet array in the absence of other components. **F** Movement of the field-free region in the assembled setup. Currents in the DC offset coils displace the position of the field-free region, with measured field data represented here in terms of the interpolated surface of points at 7.5 mT field magnitude. **G** A schematic of the micromagnetic mixing assay used to validate selective torque-based actuation. Representative examples of initial and final scans showing the distribution of fluorescent NPs are inset. Hall probe measurements showing field magnitude (above) and normalized changes in mixing parameter (below, $n$ = 3 independent experiments; means ± SD) are provided for target points at $x$ = ±5 mm in (**H**), $y$ = ±5 mm in (**I**), and $z$ = 0, ±5 mm in (**J**). "On-target" for the $x$ and $y$ axis refers to offset currents that bring the target point into alignment with the mixing chamber at the respective points. "On-target" for the $z$ axis refers exclusively to position at zero offset current. Source data are provided as a Source Data file. The graphical element depicting a mouse has been adapted from ref. 30. Reprinted with permission from AAAS.

to be in good agreement with expectation (Fig. 4E, Fig. S10). This array was then also used to observe the displacement range of the field-free point by introducing currents into the DC offset coils, depicted in Fig. 4F by the electronically controlled motion of an interpolated surface of constant magnitude at 7.5 mT. As expected from simulation (Fig. S11), approximately 1 cm of travel in each direction in the transverse plane was possible for sustained offset with this setup, ultimately limited in practice by heat dissipation of the uncooled portions of the DC coils.

Final characterization of the net magnetic fields produced by the fully assembled setup (Fig. S12) was conducted in tandem with a NP mixing experiment intended to validate spatially selective torque-based actuation. In brief, a cantilever was affixed to a micropositioner, which was used to maneuver a Hall probe inside the working volume, enabling measurements at known positions, in the presence of combined RMF and DC offsets. Keeping the same coordinate reference, a 3D printed 5 mm diameter cylindrical sample chamber sealed by clear polycarbonate sheet and filled with glycerol was placed at the end of the cantilever and introduced back into the setup at predetermined locations (see Fig. 4G). This chamber also contained a microscale cylindrical NdFeB magnet (300 µm diameter × 500 µm length) freely situated at the bottom, as well as a localized 1 µl injection of red fluorescent NPs toward the top. Unsuppressed rotational actuation of the micromagnet at conditions near its step-out frequency (40 Hz at 5 mT in pure glycerol) caused it to move rapidly throughout the well and convectively mix the NPs with the surrounding glycerol. Mixing was assessed with a Sapphire bimolecular imager (Azure Biosystems) using the software AzureSpot. RMF suppressed by a magnetostatic contribution resulted in oscillation of the micromagnet at the bottom of the well, with minimal contributions to mixing. The degree of mixing before and after actuation was assessed by examining normalized changes to the mixing index (defined as the mean divided by the standard deviation of fluorescence intensity within the chamber) before and after actuation, with representative examples of the scans provided in Fig. 4G.

To test the setup's ability to target off-center locations in each direction while maintaining the spatial selectivity of the field-free point, Hall probe measurements and mixing experiments were performed at paired displacements in the transverse plane for $x$ and $y$ = ±5 mm from the center. For each pair, the DC coil currents were suitably adjusted to bring the field-free point into alignment with the "on-target" coordinate, rendering the opposite displacement the "off-target" coordinate (Fig. 4H, I). Increased torque-driven mixing of fluorescent NPs through actuation of micromagnets was observed at the on-target locations (Fig. 4H, I). These results indicate spatially selective torque-based actuation in the mouse-scale setup, and moreover confirm the functionality of the DC offset coils. Spatial selection along the axis of the cylinder (i.e., in the $z$ direction, Fig. 4J) is accomplished through positioning rather than through electronic control of DC currents. Thus, the Hall probe characterization was repeated along this axis at three points, $z$ = −5 mm, 0 mm, and +5 mm, in order to verify similar functionality in this direction (Fig. 4J).

## Torque-focusing shows promise for improved bacterial tumor colonization in vivo

Following functional validation of the constructed mouse-scale setup, we proceeded with conducting a small in vivo trial to evaluate the potential of the torque-focusing to improve targeting of tumors in vivo. Local application of RMF to subcutaneous flank tumors has previously been shown to improve delivery of MTB into tumors[30]. However, an ideal approach requires compatibility with deep-seated targets. The aim of this study was to develop a versatile control strategy to selectively target tumors independent of their location. To demonstrate this, three study groups were defined in which different control strategies were applied to control MTB.

To perform the experiments, BalbC nude mice bearing subcutaneous breast cancer xenografts were intravenously administered with $8 \times 10^8$ far-red stained MTB (Fig. 5A). Injected mice were then put under anesthesia for 1 h while exposed to a magnetic stimulus (Fig. 5A). The mouse-scale setup was used to focus RMF (at 15 mT and 12 Hz) only on tumors in the selection field group. This was accomplished by applying DC currents tailored to the respective locations of each tumor and using a finely gridded sliding track for axial positioning (Fig. S13). The group with global RMF was placed in a separate three-phase RMF coil in the absence of any selection field. The third study group was treated without any magnetic stimulus. Subsequently, the mice were returned to their cages and their blood was sampled 24 h post intervention. These samples were later used for cytokine analysis (Fig. 5A). Two end points, 24 h and 48 h, were considered for each group allowing observation of tumor colonization with time. Similar to the spheroid experiments, magnetic actuation was followed by taxis-driven motion in this timeframe, reflecting an overall hybrid scheme for tumor colonization. Mice were sacrificed at the respective end point and their tumors were harvested for further analysis through different readouts (Fig. 5A).

Histological sectioning of tumors enabled evaluation of MTB spatial distribution inside the tumors. Tumors extracted 24 h after the intervention were used to visualize penetrated bacteria stained with the proliferative dye. Representative confocal images collected through slices from the tumors reflected differential intratumoral distribution of MTB based on the control strategy (Fig. 5B). Tumors exposed to the combination of selection field and RMF exhibited more clusters of bacteria distributed across the tumor compared to tumors treated with global RMF and control (Fig. 5B). While the difference between SF + RMF and control can be attributed to the benefits of torque-driven enhanced transport, higher MTB accumulation compared to global RMF could be explained by differences in net effective dose reaching the tumor site. Global RMF is expected to lead to higher extravasation in off-target areas, which limits the available dose for extravasation at the tumor region.

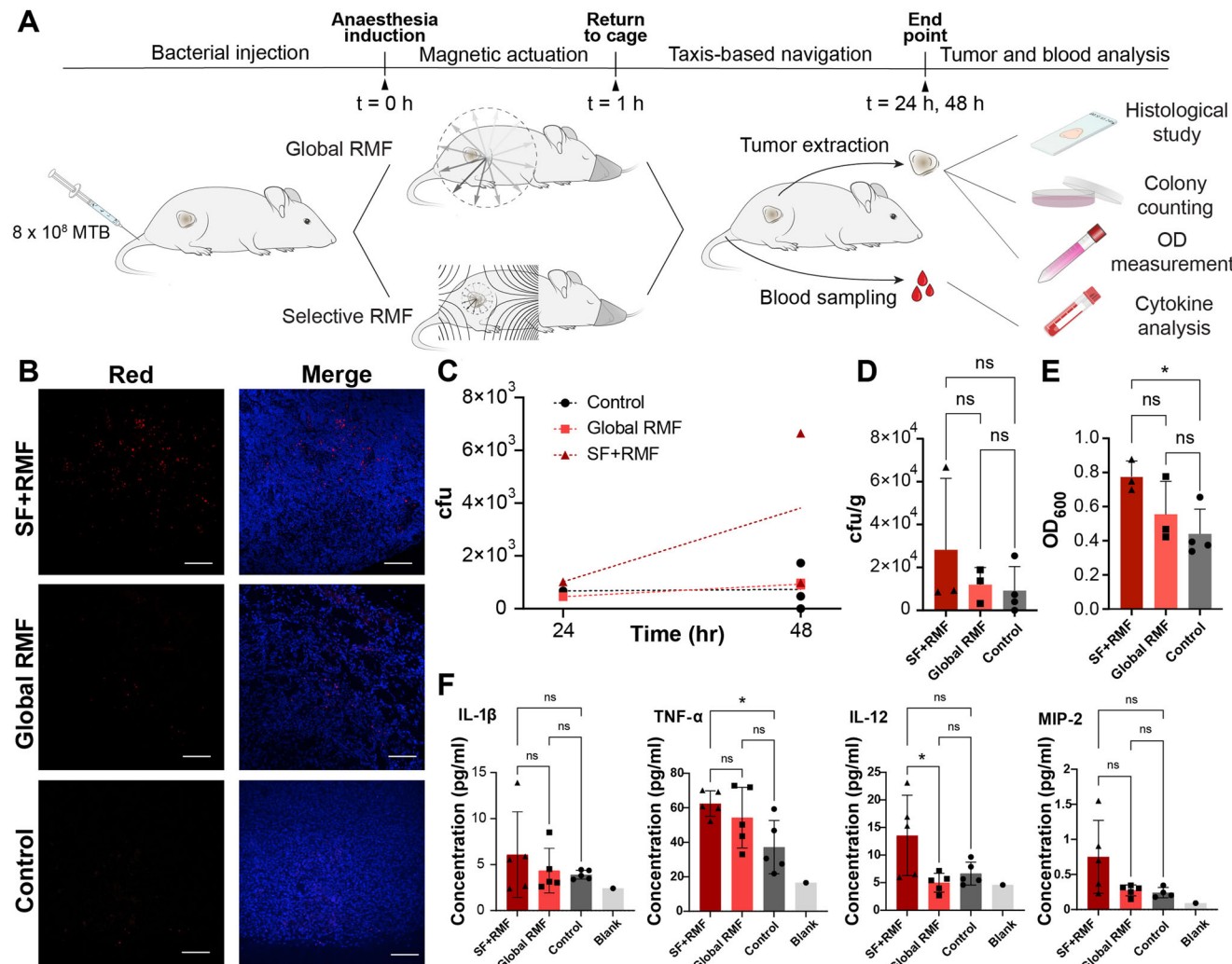

**Fig. 5 | In vivo pilot study demonstrates the promise of torque-focusing.**
**A** Timeline of the in vivo study to compare MTB tumor colonization under different magnetic stimuli. Following the injection of far-red stained MTB, mice were anaesthetized for 1 h while exposed to magnetic actuation. Blood samples were collected at 24 h. When experimental end points, 24 h or 48 h, were reached, different types of readout were utilized to assess the tumor colonization.
**B** Histological analysis of sectioned tumors. Representative confocal images illustrate higher clusters of far-red MTB specifically in deeper areas of the tumors exposed to the selection field (scale bars: 100 μm). Cell nuclei were stained with DAPI (blue). **C** Number of colony forming units (cfu) of MTB cultures on agar plates over time. Regression lines reflect higher absolute count and proliferation rate for tumor homogenates from the SF + RMF group. **D** Normalized cfu counts by tumor masses (biological replicates, independent animals; $n = 3$ SF + RMF and Global RMF, $n = 4$ Control; means ± SD; one-way ANOVA followed by Tukey's multiple comparisons test). Although not statistically significant, the SF + RMF group show 3.1 and 2.4 fold increase compared to the control and the global RMF group, respectively.

**E** Optical density of MTB cultures in liquid media (biological replicates, independent animals; $n = 3$ SF + RMF and Global RMF, $n = 4$ Control; means ± SD; *$p < 0.05$, one-way ANOVA followed by Tukey's multiple comparisons test). $OD_{600}$ was measured for each tumor homogenate following 8 days of culture. Exposure to the selection field results in a significant increase in $OD_{600}$ compared to the control ($p = 0.048$). **F** Serum cytokine levels at 24 h measured with Luminex multiplex assay (biological replicates, independent animals; $n = 5$ SF + RMF, Global RMF, and Control; $n = 1$ Blank; means ± SD; *$p < 0.05$, one-way ANOVA followed by Tukey's multiple comparisons test). Prior to statistical analysis, outliers were removed using the ROUT method with ($Q = 5\%$). The SF + RMF group exhibited higher cytokine levels in a consistent manner with the tumor accumulation data. Source data are provided as a Source Data file. Graphical elements depicting readout methods have been adapted from free vector files provided by www.vecteezy.com. Graphical elements depicting mice and a syringe have been adapted from ref. 30. Reprinted with permission from AAAS.

To further quantify tumor colonization by MTB under different conditions, most of the tumors were homogenized and cultured partly in liquid media and partly on solid agar plates. Colonies of MTB on the plates were counted 4 days post culture. Higher proliferation of MTB over time was observed in mice exposed to selective actuation, as demonstrated by the slope of regression lines for each condition (Fig. 5C). The control group was characterized with lower colony forming units (cfu) and a relatively constant density over time. This difference may indicate the presence of a critical threshold for the number of infiltrating bacteria that can support intratumoral proliferation and subsequent robust colonization. Application of the

selection field potentially lowers the minimum required dose to reach this threshold. Pooling the normalized counts per group also revealed higher, though not statistically significant, MTB colonization with application of the focused torque density (3.1 fold increase SF + RMF versus control, $p = 0.459$, 2.4 fold increase SF + RMF versus global RMF, $p = 0.596$, Fig. 5D). Under both quantification methods, global RMF yielded comparable results to the control without magnetic actuation ($p = 0.981$ Fig. 5D). This implies that although torque-based actuation improves extravasation of the magnetic microrobots, its uniform application throughout the body could undermine its benefits for tumor targeting.

Optical density measurements at 600 nm performed on the MTB cultures in liquid media confirmed the above findings. $OD_{600}$ values, following 8 days of culture, implied that significantly higher number of bacteria were present in the cultures of tumors exposed to the selection field compared to control ($p = 0.048$, Fig. 5E). Corresponding values for cultures of tumor homogenates from the global RMF group fell between the other two groups (Fig. 5E). Although the differences in the OD values between conditions do not provide a direct count of the viable MTB population within the tumors, these measurements do reflect initial seeding concentrations and subsequent proliferation. While both cfu counts and OD values assess the number of viable bacteria found in a tumor, the former is prone to higher variability, leading to different levels of statistical confidence.

Lastly, serum cytokine levels were analyzed from blood samples collected at the 24 h time point. Given the use of nude mice for this study, the focus was placed on mediators of innate immunity. The main pro-inflammatory cytokines secreted by cells such as macrophages, as part of the innate immune response, were found to be within anticipated levels in all conditions. Elevated levels of IL-1β, TNF-α, IL-6, and IL-12 were observed for mice that were exposed to selective actuation compared to control, which in most cases were not significant (Fig. 5F and fig. S14A). Some pro-inflammatory chemokines such as MIP-2 that mainly attract neutrophils also demonstrated higher levels in mice treated with the selection field (Fig. 5F and Fig. S14B). The same pattern was observed for anti-inflammatory cytokines such as IL-10 (Fig. 5F and Fig. S14C). The cytokine levels generally followed the same trend as the bacterial tumor accumulation. Previous in vivo studies with strain AMB-1 have reported similar clearance of bacteria from other organs after 24 hr, independent of magnetic stimulus. Thus, higher levels of cytokines may suggest that immune cells encounter and respond to higher bacterial loads as a result of enhanced accumulation and proliferation in tumors treated with SF.

## Discussion

It has previously been asserted that magnetic torques rather than magnetic forces are advantageous for actuating the smallest microrobots to efficiently transfer mechanical energy via externally applied magnetic fields. The present work illustrates a further advantage of torque-based schemes–their compatibility with an open-loop control strategy for spatially restricted transmission of torque density, based on combining RMFs with magnetostatic selection fields. One way to view the mouse-scale setup that was developed and validated here is as an unusual type of three-phase motor with numerous diffuse and distributed rotors, accompanied by a mechanism to exert spatial control over which of these rotors are effectively locked and which are free to rotate. One clear alternative approach to achieve selective rotational actuation is to apply spatially limited RMFs, a strategy that was indeed exploited in our previous work[30]. Nevertheless, the introduction of a selection field restricts a target point more tightly and controllably for torque application, in addition to permitting target points that are remote from the coils or electromagnets generating the RMF.

The selection field that focuses the transferred torque density is characterized by gradients. In the context of off-target suppression, these gradients help to define spatial resolution, however, they can also give rise to forces that potentially interfere with rotational actuation by drawing bacteria away from a target region. In the limit of weak gradients or microrobots with relatively small, noninteracting magnetic moments that are dominated by torque rather than force, the influence of these pulling forces is small. To assess whether this is the case for the bacteria studied here and shed light on the underlying mechanisms behind the observed effects, it is possible to perform an order-of-magnitude analysis of the forces involved in the studied actuation scheme. The gradient forces in our setups can be estimated in the range of 0.5 to 5 fN. The propulsive forces that drive taxis-driven

motion have been reported to be 0.1–0.5 pN[30]. Finally, the forces arising from rotational actuation fall within 0.5–1 pN[30]. A comparison between these values implies that the gradient-based forces play a substantially weaker role in the conducted assays relative to the other forces which are key to our actuation scheme. Moreover, the improved spatial selectivity in convection-enhanced NP transport experiments employing the mouse-scale setup reveals diminishing contribution of gradient forces at larger scales.

Although the use of selection fields in this work was inspired by the mechanism for signal isolation employed in MPI, key differences should be noted. Firstly, the resolution of field-free regions in MPI depends directly on material properties of the magnetic tracer NPs, whereas the resolution here is set primarily by the magnitude of the RMF relative to the selection field gradient. Microrobot design thus influences resolution of the transferred torque density only indirectly, because magnetic and hydrodynamic characteristics do ultimately dictate optimal RMF conditions. Secondly, in MPI, selection fields with higher gradients are almost always desirable to improve resolution, aside from the technical difficulties associated with generating them. In the case of focused torque transfer, gradients need to remain low enough to avoid exerting forces that pull microrobots against surfaces and interfere with their rotation, an effect that may have played a role in the miniaturized in vitro experiments presented here. This implies that for each microrobot design, there exists both a minimum selection field gradient needed to provide the required resolution, as well as a maximum gradient that can be tolerated before interfering with torque-based actuation. Fortunately, for the subset of microrobots small enough to be suitable for intravenous administration (typically less than 10 μm) like the MTB employed here, and for gradients that can be realistically generated at physiological scales, the influence of torque typically far exceeds that of gradient forces.

Wheras it would likely be possible to preserve the basic geometry of the in vivo setup developed here while scaling it up to the size of human patients, more efficient approaches are readily foreseeable. Our in vivo experiment specifically required an apparatus capable of exposing the whole body of a mouse to an RMF, whereas this requirement could be relaxed for a clinical instrument dedicated to focused torque application. Constructed around the working distance as a key scaling parameter, a comparatively smaller permanent magnet array could be employed, possibly with mechanical adjustability for zero-point focusing. Our in vitro experiments showed functional benefits to using swept RMFs, suggesting that an ideal clinical apparatus should also be capable of sweeping its plane of rotation. Finally, a relatively high standard deviation was observed in the selective actuation group of the in vivo study. Given spatially targeted nature of the torque-focusing, tumor colonization under this strategy is subject to higher sensitivity to alignment. As a result, compatibility with imaging modalities such as ultrasound or computed tomography in order to initially pinpoint target coordinates accurately is likely an important consideration informing the design of a clinically relevant device.

The in vivo pilot study was aimed at investigating the potential of this spatially selective open-loop control strategy while adhering to animal welfare principles by using small cohorts. Given the promise of this strategy, illustrated by the results of our preliminary in vivo study, follow-up trials with larger cohorts may further elucidate the benefits of targeted torque-driven schemes for both living and synthetic microrobots. One goal could be to find the maximum tolerated dose of MTB in long-term experiments and study how much this value is influenced by the introduction of the selection field. Also, this study was conducted at 25% lower magnetic field magnitude compared to our previous work to ensure comfortable and comparable environmental conditions for the actuated mice. The effect size of torque-focusing at different RMF magnitudes and, in turn, varied resolutions is yet to be explored. In the context of tumor treatment, further insight into aspects other than delivery, such as immune response, is needed

before using MTB as a living therapeutic vector. In an optimal scenario, achieving spatially selective delivery holds the potential to better control immune reactions. Furthermore, this effect could be fine-tuned by co-delivering pharmacological agents, thereby orchestrating a robust and targeted anti-tumor response.

The concept of focused application of torque density to biomedical microrobots could eventually be applied to numerous magnetic microrobot designs. The underlying logic of torque suppression using selection fields may even prove applicable to non-rotating torque-based locomotion schemes, such as those employing beating magnetic fields, a topic that could warrant further investigation. Many different apparatuses can also be envisioned for focused torque-based actuation, ranging from altered use cases of existing MPI systems to specialized devices that incorporate permanent magnet arrays. As an open-loop control strategy, torque-focusing is appealing because it brings together the simplicity and passivity of a defined target point with a scheme that powers the activity of biomedical microrobots.

## Methods

### MTB liquid and semi-solid culture

Wild type *Magnetospirillum magneticum* strain AMB-1 (ATCC 700264) and genetically modified AMB-1 to express GFP were used for this study. For liquid cultures, bacteria were grown in revised magnetic spirillum growth medium (MSGM; ATCC Medium 1653) containing 0.68 g of potassium phosphate (Sigma Aldrich), 0.37 g of succinic acid (Sigma Aldrich), 0.37 g of tartaric acid (Sigma Aldrich), 0.12 g of sodium nitrate (Sigma Aldrich), 0.035 g of ascorbic acid (Sigma Aldrich), 0.05 g of sodium acetate (Sigma Aldrich), 10 ml of Wolfe's Vitamin Solution (ATCC), 5 ml of Wolfe's Mineral Solution (ATCC), and 2–4 ml of 10 mM Ferric Quinate per liter of ultrapure water. pH of the media was adjusted to 6.75 by adding 1 M NaOH. Bacteria were passaged at a 1 to 10 ratio every 6–10 days by centrifuging the culture tubes at 9500 rpm for 10 min and resuspending in fresh medium. In order to provide microaerophilic condition, the optimal growth environment for this strain, 15 ml culture tubes were filled to the top leaving a small amount of headspace followed by sealing the screw cap with Parafilm before incubating at 30 °C. For semi-solid cultures, 10× MSGM and 0.7% solution of agar (Sigma Aldrich) in deionized water were prepared separately. Solutions were then autoclaved and mixed in a 1 to 9 ratio to reconstruct 1× MSGM for semi-solid media. The vitamin solution (10 μl/1 ml) and the mineral supplement (2 μl/1 ml) were added to the plates prior to solidification of the agar. Plates were stored at 4 °C before use. To provide the proper microaerophilic conditions for AMB-1 in culture, BD GasPak™ EZ Campy Pouch™ Systems comprising a small sachet and a resealable bag were used. Two petri dishes were placed in a single bag and incubated at 30 °C. MTB colonies were visible 4–6 days post-culture under these conditions.

### Microfluidic device fabrication

All microfluidic chips were fabricated using poly-dimethyl siloxane (PDMS) according to standard soft lithography protocols. Microfluidic structures including channels (for MTB-collagen studies) and chambers (for selective transport studies) incorporated the contact line pinning method and were designed in AutoCAD. Photolithography techniques were then employed to make the corresponding master molds out of 4-in. silicon wafers. Briefly, wafers were spin-coated by a ~130 μm thick layer of highly viscous negative photoresist (SU-8 3050, MicroChem). The first layer underwent the UV exposure through a foil mask containing patterns without the contact lines. Following sufficient baking and subsequent resting at the room temperature, a low viscosity negative photoresist (SU-8 3010, MicroChem) was used to deposit a thin second layer (~20 μm) atop the wafer. UV treatment was performed by placing the second foil mask featuring the contact lines while ensuring proper alignment of the layers. This step was followed by baking the photoresist, developing the unexposed areas, and

silanization of the wafer with chlorotrimethylsilane (Sigma Aldrich) to prepare the master mold for device fabrications.

Microfluidic chips were made by mixing PDMS elastomer base and curing agent (SYLGARD 184 Silicone Elastomer Kit, Dow Corning) in 1:10 ratio by weight. The mixture was fully degassed and then poured on top of the silicon master, followed by another brief degassing step. The bubble-free mixture was placed inside a 70 °C oven for at least 3 h. Later, the cured PDMS slab was peeled off the mold and cut into individual devices which were then punched for inlets and outlets. 1 mm biopsy puncher (Kai Medical) was used for all the connections unless otherwise mentioned. Lastly, devices were bonded to microscope slides using air plasma for 50 s followed by a brief baking at 80 °C.

For spheroid colonization experiments, five-chamber devices were only punched at the center of each well with 3 mm punchers resulting in five chambers for placing the spheroids. For fluorescence imaging of spheroids on the following day, PDMS wells were made using a blank silicon wafer. The resulting plain PDMS slab from this wafer was punched concentrically with 6 and 12 mm punchers. The cut ring-shaped pieces were bonded to coverslips and used later for imaging.

### Computational modeling of magnetic actuation and NP transport

The Magnetic Fields No Currents physics interface of COMSOL Multiphysics was utilized to model the magnetic field distribution inside the working space of the microfluidic devices. Four blocks representing the respective permanent magnets were created according to their experimental layout. Size of the blocks and their mutual distances were varied in a parametric study. Each magnet was modeled as a permanent magnet possessing a remnant flux density $|\mathbf{B_r}|$ of 1.4 T and a recoil permeability $\mu_{rec}$ of 1.05.

To study NP transport, the geometry of a single well from the multi-chamber design was recreated. Governing equations from Stokes Flow and Transport of Diluted Species interfaces were solved for fluid velocity and concentration of the particles. The effect of magnetic actuation on MTB was modeled through application of a volume force corresponding to the torque-driven flow of bacteria in such chambers. The diffusion coefficient of the NP in liquid was calculated from the Stokes–Einstein relation ($D_0 = 2 \times 10^{-12}$ m²/s). Collagen gel was modeled as a porous material with porosity $\varepsilon$ of 0.6. Hydraulic permeability was assumed to be $\kappa = 10^{-16}$ m² and diffusion coefficient was calculated by modifying the fluid diffusion coefficient according to the tortuosity model for effective diffusivity.

### Selective actuation of MTB for NP transport

An experimental platform featuring five separate microfluidic wells (North, East, South, West, Target) made from PDMS was fabricated according to standard soft lithography protocols. Each well featured a circular ridge to allow surface tension-based contact line pinning of viscous collagen (Fig. S2). 2–3 days after plasma bonding, the microfluidic devices were fitted to hold four permanent magnets (top, bottom, left, right) within precut slots for accurate positioning and creation of a field-free region. To generate a desired selection field with the zero point in the center well (target), a simplified version of the Halbach cylinder[50] $k = 2$ was adopted using NdFeB block magnets (6 × 4 × 2 mm or 5 × 2.5 × 1.5 mm, N45, Supermagnete.ch). At the top and the bottom, arbitrarily defined north poles faced each other while north poles pointed outward on the left and the right sides. Upon assembly, collagen was filled into the central part of each well. Following 45 min incubation at 37 °C, the devices were stored in humidified petri-dishes at room temperature before the start of the experiment. MTB were spun down and resuspended in phosphate-buffered saline (PBS, Sigma Aldrich) at a final concentration of $8.7 \times 10^9$ cells/ml. To quantify NP transport, 1 μl of red fluorescent NPs

(FluoSpheres carboxylated microspheres, 0.2 μm) were added per 100 μl of the MTB suspension. The final suspension was then introduced into the annular part of the wells surrounding the central collagen compartment, followed by torque-based magnetic actuation for 1 h. Transport evolution was monitored and captured by confocal micrographs at $t = 0$, $t = 30$ min and $t = 1$ h.

Image processing was performed in ImageJ (NIH). The collagen area was determined via the Analyze Particles command applied to the binarized image of the red channel. Fluorescent signal from NPs inside this ROI was integrated for different time points. The values of later time points were normalized by the time point 0 to compensate for any initial penetration of the NPs caused by the filling process. The ROI defined by the collagen was shrunk to form bands at certain intervals which enabled quantification of the signal as a function of the distance from the interface.

## Magnetic actuation in small proof-of-concept setup

The RMF was produced in these experiments (shown in Figs. 2 and 3) with a commercial electromagnet setup consisting of eight coils forming a hemisphere at one end as the working space (MFG-100-i, Magnebotix AG). The setup was integrated into an inverted microscope (Nikon Ti Eclipse) with a spinning disc confocal scanner unit (CSU-W1, Yokogawa), controlled by NIS-elements AR version 5.02.03. This enabled live imaging of the samples under magnetic actuation. The system can generate uniform arbitrary 3D magnetic fields within 1 cm³ in space. Sweeping RMF was applied through application of the following input functions:

$$B_x = B \sin(2\pi f_2 t) \sin(2\pi f_1 t) \tag{1}$$

$$B_y = B \cos(2\pi f_2 t) \sin(2\pi f_1 t) \tag{2}$$

$$B_z = B \cos(2\pi f_1 t) \tag{3}$$

Where $f_1$ is the frequency of the out-of-plane RMF and $f_2$ represents the sweeping frequency of the plane of rotation. In order for the plane of rotation to go through one revolution during the actuation, $f_2$ was set to 1/3600. Under the especial case of $f_2 = 0$, the actuation scheme turns into constant RMF with $yz$ as the plane of rotation. For experiments with the alternating RMF, one plane of rotation, i.e., $xz$, with switching axis of rotation between $y$ and $-y$ directions was used.

## Spheroid culture

MCF-7 cells (ATCC, HTB-22) were cultured in high glucose Dulbecco's Modified Eagle's Medium (DMEM, ThermoFischer) supplemented with 10% fetal bovine serum (FBS, BioWest) and 1% Penicillin/streptomycin (P/S, Corning). Ultralow adhesion plates (InSphero) were utilized to form tumor spheroids from MCF-7 cells. A seeding density of 10,000 cells/well in 50 μl of growth media was used in 96 well plates. The well plates were centrifuged at $500 \times g$ for 10 min followed by incubation at 37 °C with 5% CO₂. The size of the tumor spheroids reached ~400 μm after 3 days. DNA staining with Hoechst 33342 at a final concentration of 5 μg/ml in media was performed before the experiment for 1 h at 37 °C.

## In vitro spheroid colonization under torque-focusing

To stain the bacteria and allow tracking of subsequent generations, 2 μl of a far-red proliferative dye (CellTrace™ Far Red Cell Proliferation Kit, ThermoFischer) was added to 1 ml of bacteria suspension at $5 \times 10^8$ cells/ml, assessed by optical density measurements. Following 20 min of agitation on a shaker while protected from light, the dye was deactivated using 100 μl of DMEM for 10 min. The bacteria were spun down and resuspended in 1 ml DMEM for subsequent experiments.

For these experiments, the five-chamber devices were punched at the center of each well with 3 mm punchers resulting in five wells for placing the spheroids. Corresponding slots for small magnets were cut and magnets corresponding to the selection field with lower gradient ($5 \times 2.5 \times 1.5$ mm) were placed following the above-mentioned order for magnetization directions. Wells were rinsed with ethanol, air dried, and then exposed to UV for an hour for sterilization. Each well was filled with ~25 μl of MTB suspension, followed by careful release of spheroids and settling at the bottom of the wells. The devices were then covered with a lid and mounted on the microscope holder to minimize contact with the surrounding air and avoid potential contaminations. Magnetic actuation at 20 mT and 14 Hz was applied for 1 h, followed by collection of the spheroids, thorough washing with media, and incubation in sterile well plates at 37 °C and 5% CO₂ for 24 h.

Spheroids were transferred into PDMS pools, bonded on thin cover slips for high-resolution confocal imaging after 24 h. Images were taken along the Z-axis for a distance of 200 μm at 10 μm intervals. To quantify the accumulation of bacteria in tumor spheroids, the ROI was defined as the outer edge of the spheroid in a binarized Z-projected image of the DAPI channel in ImageJ. The far-red signal was then integrated for all planes according to the respective ROI.

## In vivo tumor model experiments

Animal experiments were conducted under license number ZH151/2020 in compliance with the regulations from Veterinäramt Kanton Zürich. Animals were housed in cages with a HEPA-filtered air supply, with the air in the room also separately HEPA-filtered and maintained at an ambient temperature of $22 \pm 2$ °C and a relative humidity of $55 \pm 10\%$. The specific temperature and relative humidity experienced in the cages depend on the number of animals (always ≥2 and ≤5), but does not deviate dramatically from these values. The animals were maintained on a 12 h light/dark cycle with 10 min transitional fades. The experiments were performed during regular working hours, which coincided with the light part of the cycle. Following 6 weeks of acclimatization, female BALB/c nude mice (Charles River) were inoculated each with $5 \times 10^6$ GFP MDA-MB-231 cells (GenTarget, SC040-Puro) in 100 μl of cell suspension in 8 mg/ml Matrigel (Corning) in the hind flank. Health status and tumor sizes were monitored for three weeks until all tumor diameters exceeded 5 mm. Subsequently, mice were randomly divided into different study groups. Except for the negative control (blank), mice were intravenously injected with $8 \times 10^8$ of far-red stained MTB in 100 μl of PBS via the tail vein. Injected mice were anaesthetized for 1 h while exposed to either no magnetic actuation (Control), globally applied rotating field (Global RMF), or spatially selective actuation with the selection field (SF + RMF). When the selective actuation was used, a DC offset field was applied to move the zero point to the tumor's location which was determined through a fine grid on the sliding track. Magnetic actuation was performed at 15 mT and 12 Hz. Following the intervention, mice were returned to their cages and monitored for weight loss and any other sign of changes in behavior. Blood samples were taken from the tail vein after 24 h and stored frozen. Before analysis, samples were centrifuged at $1000\,g$ for 10 min at 4 °C. Milliplex Mouse High Sensitivity T Cell (MHSTCMAG-70KPMX, Millipore) was used to analyze the supernatant in accordance with the manufacturer's instructions.

When the experimental end points (24 h or 48 h) were reached, mice were euthanized, and tumors and major organs were harvested for analysis. One tumor per study group was fixed in 4% paraformaldehyde and stored in 70% ethanol until processing for histology. Tumors were embedded in paraffin and 5 μm sections were cut using a microtome. The sections were stained with DAPI and the slides were then mounted and imaged. The rest of the collected tumors were weighed and then dissociated using a Tumor Dissociation Kit Human and a gentleMACS tissue dissociator (Miltenyi Biotec). Tumor homogenates were split into two parts and cultured in liquid medium or on

semi-solid agar plates. Colonies grown on agar plates were counted 4 days post inoculation. To quantify MTB in the liquid cultures, bacteria were spun down on day 8, resuspended in 1 ml of media and $OD_{600}$ was measured with a Tecan Spark plate reader.

## Statistics and data analysis

All reported values in plots represent mean ± SD of at least three replicates. Multiple unpaired *t* tests were used for statistical comparison between the two groups in the in vitro assays. For statistical analysis of the in vivo study groups, one-way analysis of variance (ANOVA) with Tukey's follow-up test was used. GraphPad Prism 8.0 software was utilized to plot graphs and perform statistical analysis. Other software used to analyze data include MATLAB 2022b, COMSOL Multiphysics 6, Fiji ImageJ, and Excel.

## Reporting summary

Further information on research design is available in the Nature Portfolio Reporting Summary linked to this article.

## Data availability

All data supporting the findings of this study are available within the article and its supplementary files. Any additional requests for information can be directed to, and will be fulfilled by, the corresponding author. Source data are provided with this paper.

## Code availability

Any scripts used for processing the raw data and visualizing the results (e.g., MATLAB, Fiji ImageJ) can be made available upon request to the corresponding author.

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

## Acknowledgements

We gratefully acknowledge the ETH Grants program for funding (ETH-23 30-2). We also thank the staff of the EPIC Animal Facility and ETH's animal welfare officers, especially Dr. Thomas Weber and Susanne Freedrich, for their support of our in vivo experimentation. Histology support was provided by Ines Kleiber-Schaaf of the UZH Center for Preclinical Development. Dr. Daphne O. Asgeirsson collected the scanning electron micrographs shown in the supplementary materials. Hanif Molayi provided support for the 3D CAD model of the mouse-scale setup. We thank the Scientific Center for Optical and Electron Microscopy (ScopeM) at ETH Zürich for access to their facilities. Dr. Cameron Forbrigger provided proofreading and discussion of an early version of the manuscript and Sonia Monti assisted with some graphical elements of the illustrations.

## Author contributions

N.M., M.G.C. and S.S. conceived the idea. N.M., M.G.C., T.G. and S.M. performed the experiments and collected data. N.M. conducted numerical simulations and visualizations. N.M., M.G.C. and S.S. wrote the original draft. N.M., M.G.C., T.G., S.M. and S.S. reviewed and edited the manuscript. S.S. supervised the project.

## Funding

## Competing interests

S.S. is a co-founder, technical advisor, and member of the board of MagnebotiX AG. All other authors declare that they have no competing interests.
