## [Peer Review File · Nature Communications]

Spatially selective delivery of living magnetic microrobots through torque-focusingREVIEWER COMMENTS

Reviewer #1 (Remarks to the Author):

This study reported an effective rotating magnetic field (RMF)-focusing and suppression off-target actuation method for targeting deeply located tumors as a magnetic field-driven microrobot challenge. As a microrobot, a magnetotactic bacteria (MTB) is not special because many researchers have used them. However, the proposed electromagnetic navigation system based on the principle of magnetic particle imaging (MPI) system is interesting in that it can target microrobots to selective positions within the workspace. In particular, the authors designed and fabricated a mouse-scale torque focusing device for in vivo pilot testing. This device was able to provide selective RMF by creating field free points in the x-, y-, and z-axes directions through magnet arrays, DC offset coils, and RMF coils. Furthermore, the authors demonstrated the targeting and potential therapeutic effect of MTB to mouse tumors. However, in the current format of the manuscript, some parts do not fully support the advantages of the proposed spatially selective RMF control. The authors need to convincingly address before the reviewer recommends publication:

1. The authors noted in Fig. 2G that nanoparticle transport into the off-target chambers was reduced when the selection field and sweeping RMF were simultaneously applied. And for this reason, it is said that it is caused by the nanoparticle being attracted by the strong gradient magnetic field generated around the target. Can you provide an approximate magnetic force applied to the nanoparticles around the target here? In addition, what strength of magnetic force is exerted on the nanoparticle in each non-target region? These data may provide useful information to the reader.
2. In the tests of Fig. 2 and Fig. 3, the target was formed only in the center of the workspace. Can the authors target NP or MTB by generating these field free points and selective RMF at points other than the center? To prove the novelty of the study, the authors need to clearly demonstrate the targeting effect using selective RMF at peripheral points including the center of the workspace in vitro and in vivo tests.
3. For readers to see the results clearly, the reviewer recommends increasing the font size of images and graphs included in all figures.
4. As in Fig 2B, the reviewer recommends that the authors display captions for blue and red in Fig. 3E to help readers' understanding.

5. Fig. 2e and Fig. 4G were not displayed in the manuscript. In addition, the results within the figure are not specified in the order in the manuscript. The reviewer took a long time to read and understand the manuscript due to the inconsistent order of the figures included in the sentence. Therefore, authors should check that all figures are displayed in the results describing them.

6. In the first paragraph of the Manuscript Section 'Application of focused torque improves tumor colonization in vivo', 'cite' should be replaced with an appropriate reference.

7. How did the authors generate field free point-based selective RMF accurately at the location of mouse tumors in vivo? In order to generate selective RMF on a tumor, imaging data such as MR, CT, X-ray, and CT to determine the location of the tumor seem to be essential. If the in vivo test was performed assuming that the tumor is located in the center of the device, this experiment may cause a large targeting error because the selective RMF was not generated at the accurate position. Therefore, if the in vivo test was performed with this assumption, it is necessary to verify the selective targeting effect by the selective RMF at various locations through the in vitro test (Fig. 2 and 3).

8. Why did the authors not clearly see the tumor treatment effect in the in vivo test? The reviewer recommends investigating changes in tumor size in the long-term (about 1 month). However, depending on the situation, it seems that it can be replaced by explaining the expected in vivo effect in the discussion.

Reviewer #2 (Remarks to the Author):

Dear Nima Mirkhani, Simone Schuerle and co-authors,

Congratulations for these nice results and very well written manuscript. You demonstrated an approach on how to use an RMF in combination with a gradient field featuring a field free region to locally select the region of interest. You were using MTB as a model microrobot and showed their behavior in vitro and in vivo, for the latter you build a dedicated setup able to accommodate mice.

I have some comments and suggestions, which you can find below as well as some minor corrections within the manuscript.

4T/m or 8T/m, respectively are very high gradient strength... I am wondering why you don't observe a magnetic force on the NP and MTBs at the off target regions. Can you comment on this?

It's still not clear to me, why the RMF and selection field leads to a higher concentration of MTB in the tumor cite compared to only RMF. I do agree that due to the combination of these fields, you can choose in which region the MTB are being actuated. Hence, it makes sense to me that you would find more MTB in a tumor inside the field free region than outside, but why are there more MTB in the tumor compared to only using RMF? You also state that these experiments were not significant, but still you argue that you observe an increase. If you don't have statistically significant results, then I would like to see at least a profound explanation.

I have some concerns about terms used throughout the manuscript, which I would like to address and ask the authors to either change it accordingly or explain why they chose these terms:

1. Magnetic gating field. I did not find where this expression is typically used. To my knowledge these kinds of magnetic fields are called gradient fields showing a field free region/point. You sometimes also name it selection field, which is also a common expression. Why you name it differently is not clear to me. And be consistent, you use all three expressions.
2. Focal torque delivery. I am struggling a bit with the expression "delivery" as it is commonly used for e.g. drug delivery. Hence, I would see terms like "application" more appropriate. Also, the expression of "focal point" is strictly/physically seen incorrect or at least misleading since a magnetic field cannot have a focus point.
3. Open loop control. Since you even use this term in the title, I was expecting to see some magnetic path guidance within the manuscript. Since, this is not what the article is about I would suggest replacing this expression. My suggestion "Spatially selective delivery of magnetic microrobots" you also don't show any drug delivery. You could add "to tumor sites" if you like.

My recommendation: accept, after revision.

Reviewer #3 (Remarks to the Author):

In this manuscript, the authors present an actuation scheme based on the combination of a rotating magnetic field typically used for the actuation of artificial bacterial flagella that the authors already used for magnetotactic bacteria, with magnetostatic gating fields. The manuscript is rather technical, the flow is difficult to follow, at least initially, and the authors change the way they present the results between the in vitro and the in vivo, such that it is uncertain that the in vivo treatment is really significantly successful.

Another main point is that the proof of concept of this actuation scheme has already been demonstrated by others in my opinion (see below), and here it rather appears as a follow-up study of the papers the group of authors published recently (DOI: 10.1126/scirobotics.abo0665).

The final main point is the fact that with this method, the authors can effectively drive AMB-1 and enhance particle diffusion locally but by using the torque driven strategy, and specifically by alternating the torque sign, the chemical and aerotactic stimuli are most likely disregarded because the torque driven forces overrule. While the MTB benefits from having local hypoxic regions around the tumor, this will remain without contribution for the particular technique. The presence of the field free region with relative high field gradients at the borders would indeed most likely always inhibit any aerotaxis or chemotaxis. A discussion around these points would be valuable for future design improvements. Does one need to resign to these taxis to have full control or is it possible to combine them and how compatible is it with the design?

Finally, there numerous small flaws that should be modified before publication can be granted. Please find below these more specific points below:

- L14-15, magnetotactic bacteria are described as a biohybrid robotic system. Why not simply a biological one?
- L18: *in vitro* is not written in italics. I am uncertain of the guidelines of the journal, but it is not in italics in the text, but in italics in the title and in the captions.
- L47: “deep tissue”, should be “deep tissue penetration” or “targeting”?
- L60-62: the concept of MPI for not only imaging but also controlling the movement of milli and microsystems is not new. It has already been demonstrated, see for example: <https://doi.org/10.1038/s41598-021-93323-4> or <https://pubs.acs.org/doi/full/10.1021/acsnm.1c00768>
- L71: at this stage, the “companion nanoparticles” were not introduced and they are a surprise that is a bit hard to understand.
- L81: “a small *in vivo* study”, what is a “small” *in vivo* study? What is the interest of a “small *in vivo* study”?
- L165 “green fluorescent NPs”, in the M&M section (l619-620), the NPs are supposed to be red fluorescent. What is right?
- L200-201 “more than 40 % ... the target region”, why are the error bars so big? Looking at Figure 2G, there is a considerable difference of NP concentration in the target area between +Sweeping RMF - Selection field and +Sweeping RMF +Selection field. Can the author comment on that?
- L224-225, figure 2F, it is unclear if the field used is with a constant RMF or a sweeping RMF? The authors should add an example for each type of RMF that they used to make the graph in Figure 2G.
- L230 “only a non significant reduction in the target chambers is observed”, it may indeed be not statistically significant but still considerable especially given the length of the error bars.
- L233 “model of cancer”, a spheroid is not a model of cancer but a model of a solid tumor, as the authors state later on in the text.
- Fig 3D, can the authors give access to their data as it is difficult to understand how the results +Sweeping RMF -Selection field and +Sweeping RMF +Selection field are significantly different with the error bars depicted in the ++ case.

- L 383, the title is misleading; the authors say the difference is not statistically significant in the text. Therefore, they should not do such a statement in the title of the section (see L421-424).
- L387 “cite”, probably a citation is missing here.
- L 408-411, In the images the authors show in Figure 5B, MTB seem to appear at deeper areas of the tumour in the global RMF group compared to the SF+RMF group. Maybe more images should be provided to prove the authors’ statement.
- L 414-416: how many experiments were performed here? No error bars are presented on Fig 5C.
- L442-444: Do the authors have a control with mice without MTB but with an applied magnetic field?
- L450-452: if the MTB are really targeting the tumour and not the surroundings, there should be a lower immune response because less MTB are around in the blood flow, can the authors explained their statement again?
- Figure 5F: the authors should be consistent with the previous graphs and add the * instead of the numerical value. This way, it would be easier for the reader to see, which differences are significant or not.
- L553-554: the bacteria are passaged by centrifugation of a culture and putting fresh medium. This is an unconventional way to grow magnetotactic bacteria, for which a 1 to 10% inoculum is typically used. Can the authors explain why they used this peculiar approach to cell culture?
- L619-621: why not using fluorescently-labeled bacteria directly?

We thank all reviewers for their insightful comments and for the time and effort they put into reviewing our manuscript. In this response document, reviewer comments are reproduced in **black** and our responses are noted in **green**.

Reviewer #1 comments:

This study reported an effective rotating magnetic field (RMF)-focusing and suppression off-target actuation method for targeting deeply located tumors as a magnetic field-driven microrobot challenge. As a microrobot, a magnetotactic bacteria (MTB) is not special because many researchers have used them. However, the proposed electromagnetic navigation system based on the principle of magnetic particle imaging (MPI) system is interesting in that it can target microrobots to selective positions within the workspace. In particular, the authors designed and fabricated a mouse-scale torque focusing device for in vivo pilot testing. This device was able to provide selective RMF by creating field free points in the x-, y-, and z-axes directions through magnet arrays, DC offset coils, and RMF coils. Furthermore, the authors demonstrated the targeting and potential therapeutic effect of MTB to mouse tumors. However, in the current format of the manuscript, some parts do not fully support the advantages of the proposed spatially selective RMF control. The authors need to convincingly address before the reviewer recommends publication:

Response:

We would like to express our gratitude towards the reviewer for their valuable and insightful comments, which have helped us improve the quality of our manuscript. Please find our detailed response to each comment below.

1. The authors noted in Fig. 2G that nanoparticle transport into the off-target chambers was reduced when the selection field and sweeping RMF were simultaneously applied. And for this reason, it is said that it is caused by the nanoparticle being attracted by the strong gradient magnetic field generated around the target. Can you provide an approximate magnetic force applied to the nanoparticles around the target here? In addition, what strength of magnetic force is exerted on the nanoparticle in each non-target region? These data may provide useful information to the reader.

Response:

We thank the reviewer for this comment and see that treatment of this matter should be made clearer. Firstly, there seems to have been a slight misunderstanding regarding the underlying cause of off-target suppression. Off-target suppression comes from vector addition of the RMF and magnetostatic selection field (Fig. 1B), which extinguishes the rotational character of the field, and thus, the efficiency of torque transfer, in regions outside the field-free point. The NPs in the experiment that is referenced (Fig. 2G) were not magnetic, and thus, were not affected directly by any appreciable magnetic gradients. However, as demonstrated in one of our previous studies (<https://www.science.org/doi/full/10.1126/sciadv.aav4803>), exposing MTB mixed with non-magnetic NPs to uniform RMF causes their convective transport. Therefore, any NP accumulation in the collagen—beyond what can be attributed to diffusion (control conditions)—can be interpreted a result of hydrodynamic coupling under magnetic actuation of MTB. While RMF alone (both with constant and sweeping planes) enhances the transport in all chambers, the selection field extinguishes this torque-driven transport in off-target areas by pinning the magnetic moment of MTB, preventing them from rotating fully, and thus, from exerting torque efficiently. The gradient caused by the selection field should ideally be small enough that it does not interfere with the main mode of transport, i.e. magnetic torque-based actuation. Given an upper bound of the gradient of 10 T/m in our setup, an estimate of resulting gradient forces can be made using the average dipole moment of MTB (<https://onlinelibrary.wiley.com/doi/abs/10.1002/adfm.202003912>) as follows:

$$F = \nabla(\vec{m} \cdot \vec{B}) \approx 5 \times 10^{-16} \text{ A m}^2 \times 10 \frac{\text{T}}{\text{m}} \approx 5 \times 10^{-15} \text{ N} \equiv 0.005 \text{ pN}$$

The flagellar propulsive force has been estimated to be 0.1-0.5 pN, with forces arising from the RMF-induced rotation falling within 0.5-1.0 pN. As a result, the contribution of gradient forces is notably weaker than the intrinsic bacterial and external torque-driven forces. Even so, such gradient forces could lead to 100s of μm displacement within an hour. Since this slow displacement process opposes transport towards the target region in our miniaturized setup (primarily outward forces away from the center), this configuration serves as a stringent test for our actuation scheme.

The following change has been made to the main text:

“One possible source of the reduction in transport at the target point is the role of gradient forces within the experimental setup. Whereas the magnetostatic field applied here is intended primarily to extinguish torque density, its associated gradients can also apply forces to MTB that draw them to the boundary of the chambers and interfere with rotational actuation. *The respective average magnitudes of the relevant forces can be estimated to be 0.5 pN for torque based forces³⁰, 0.2 pN for intrinsic bacterial propulsive forces³⁰, and 0.005 pN for magnetic gradient forces (given an upper bound of 10 T/m). Since the gradients in this miniaturized proof-of-concept setup are approximately twice that of the highest gradient realized in the larger setup described in a later section, these forces, while comparatively small, are expected to play a more dominant role here.*”

And a more general explanation in the discussion section is added:

“The selection field that focuses the transferred torque density is characterized by gradients. In the context of off-target suppression, these gradients help to define spatial resolution, however they can also give rise to forces that potentially interfere with rotational actuation by drawing bacteria away from a target region. In the limit of weak gradients or microrobots with relatively small, noninteracting magnetic moments that are dominated by torque rather than force, the influence of these pulling forces is small. To assess whether this is the case for the bacteria studied here and shed light on the underlying mechanisms behind the observed effects, it is possible to perform an order-of-magnitude analysis of the forces involved in the studied actuation scheme. The gradient forces in our setups can be estimated in the range of 0.5 to 5 fN. The propulsive forces that drive taxis-driven motion have been reported to be 0.1-0.5 pN.³⁰ And lastly, the forces arising from rotational actuation fall within 0.5-1 pN.³⁰ A comparison between these values implies that the gradient-based forces play a substantially weaker role in the conducted assays relative to the other forces which are key to our actuation scheme. Moreover, the improved spatial selectivity in convection-enhanced NP transport experiments employing the mouse-scale setup reveals diminishing contribution of gradient forces at larger scales.”

2. In the tests of Fig. 2 and Fig. 3, the target was formed only in the center of the workspace. Can the authors target NP or MTB by generating these field-free points and selective RMF at points other than the center? To prove the novelty of the study, the authors need to clearly demonstrate the targeting effect using selective RMF at peripheral points including the center of the workspace in vitro and in vivo tests.

Response:

We share the interest of the reviewer in establishing our ability to move the field-free point. Indeed, we used DC offset fields for this purpose both in vitro (shown in Fig. 4 F, G, and H) and in vivo (Fig. 5) with our custom mouse-scale setup to displace the field-free point, and thus, target regions away from its center. To emphasize this important point, we have adapted the relevant text for both *in vitro* and *in vivo* tests:

“To test the setup’s ability to target off-center locations in each direction while maintaining the spatial selectivity of the field-free point, Hall probe measurements and mixing experiments were performed at paired displacements in the transverse plane for x and $y = \pm 5$ mm from the field-free point.”

“The mouse-scale setup was used to focus RMF (at 15 mT and 12 Hz) only on tumors in the selection field group. This was accomplished by applying DC currents tailored to the respective locations of each tumor and using a finely grided sliding track for axial positioning.”

The experiments presented in Fig. 2 and Fig. 3 were intended as proof-of-concept demonstrations of selective RMF, and for that, we utilized a commercially available electromagnet setup primarily designed for micro-manipulation without a DC offset coil. Here, we focused first on the fundamental concept and theoretical aspects of torque focusing, and later introduced the DC offset for target displacement by constructing a custom setup that could also help us in investigating scalability and translatability to in vivo experiments.

3. For readers to see the results clearly, the reviewer recommends increasing the font size of images and graphs included in all figures.

Response:

To address this concern, we increased and harmonized the font size across all five figures.

4. As in Fig 2B, the reviewer recommends that the authors display captions for blue and red in Fig. 3E to help readers' understanding.

Response:

We thank the reviewer for noting this possibility to improve Fig. 3. We have accordingly incorporated this change, reproduced below for convenience.

5. Fig. 2e and Fig. 4G were not displayed in the manuscript. In addition, the results within the figure are not specified in the order in the manuscript. The reviewer took a long time to read and understand the manuscript due to the inconsistent order of the figures included in the sentence. Therefore, authors should check that all figures are displayed in the results describing them.

Response:

We are sorry for any inconvenience caused by this. We believe that the reviewer refers to Fig. 2E and Fig. 4J which are now referenced in the text. Additionally, we carefully reviewed the updated version to ensure that all figures are properly first referenced in chronological order and described in the text.

“This effect should be mitigated by continuously varying the direction of rotation, and more uniform enhanced convection throughout the device is also expected to modestly increase overall NP accumulation in the collagen (Fig. 2E).”

“These results indicate spatially selective torque-based actuation in the mouse-scale setup, and moreover confirm the functionality of the DC offset coils. Spatial selection along the axis of the cylinder (i.e. in the z direction, Fig. 4J) is accomplished through positioning rather than through electronic control of DC currents.”

6. In the first paragraph of the Manuscript Section ‘Application of focused torque improves tumor colonization in vivo’, ‘cite’ should be replaced with an appropriate reference.

Response:

Thank you for bringing this mistake to our attention. This has been fixed in the revised version:

“Local application of RMF to subcutaneous flank tumors has previously been shown to improve delivery of MTB into tumors.³⁰”

7. How did the authors generate field free point-based selective RMF accurately at the location of mouse tumors in vivo? In order to generate selective RMF on a tumor, imaging data such as MR, CT, X-ray, and CT to determine the location of the tumor seem to be essential. If the in vivo test was performed assuming that the tumor is located in the center of the device, this experiment may cause a large targeting error because the selective RMF was not generated at the accurate position. Therefore, if the in vivo test was performed with this assumption, it is necessary to verify the selective targeting effect by the selective RMF at various locations through the in vitro test (Fig. 2 and 3).

Response:

We fully agree with the reviewer regarding the necessity of integrating an imaging system with the setup, especially in the context of clinical applications and deep-seated tumors. As previously addressed in response to comment 2, we indeed employed DC offset fields to move the field-free point in both our in vitro (Fig. 4) and in vivo (Fig. 5) experiments. For the present study, our focus was directed towards elucidating the actuation mechanism and system design. This led us to opt for subcutaneous flank tumors as a simple model, where the tumor locations could be determined without the need for an imaging modality. To ensure precise alignment of the field-free point with the tumor locations and minimize targeting errors, we used a finely gridded sliding track for both positioning the mouse and performing our calibration measurements. In other words, knowing the location of the tumor by reading its coordinates on the sliding track enabled us to apply the corresponding DC offset to accurately target the off-center tumor locations within the setup.

We have clarified this in the methods and added an image of the sliding track in the Supplementary Information, reproduced below for the reader’s convenience.

Fig S12: Finely gridded sliding track for axial positioning of tumors. A gridded track made from laser cut acrylic sheets was used to position mice with tumors within the setup in the in vivo study.

8. Why did the authors not clearly see the tumor treatment effect in the in vivo test? The reviewer recommends investigating changes in tumor size in the long-term (about 1 month). However, depending on the situation, it seems that it can be replaced by explaining the expected in vivo effect in the discussion.

Response:

The in vivo experiment described had the primary purpose of assessing the influence of our selective actuation scheme on the delivery of MTB, which defines the scope of this manuscript. To observe therapeutic effects, a longer time span of observation and the addition of therapeutic cargo would have been needed, which is something we indeed are investigating in other work (<https://www.biorxiv.org/content/10.1101/2023.03.31.535049v1.abstract>). Here, we only considered acute effects on immune response, which is related to the delivery efficiency. Generally, sufficient delivery of bacteria to a tumor is a crucial first step towards effective tumor treatment and therefore a worthwhile study goal on its own. Bacterial cancer therapy is a rich field with numerous entire studies devoted to understanding and demonstrating therapeutic effects. In addition, in most cases, the immune-mediated impact of bacteria is supplemented by using them as carriers for therapeutic cargo, which is either produced by genetically engineered bacteria or conjugated to their surface via membrane functionalization. The possibility also exists to functionalize non-magnetic bacteria with magnetic materials to enable magnetic control, another active field of study. We ultimately aim to broadly support various types of bacterial cancer therapies (and even beyond to non-living magnetic agents) with our actuation scheme, which is why we limited the scope in this way.

Nevertheless, in response to the reviewer's suggestion about discussing the expected effects in the manuscript, we have added the following to the discussion to better address this in the revised version:

“In the context of tumor treatment, further insight into aspects other than delivery, such as immune response, is needed before using MTB as a living therapeutic vector. In an optimal scenario, achieving spatially selective delivery holds the potential to better control immune reactions. Furthermore, this effect could be fine-tuned by co-delivering pharmacological agents, thereby orchestrating a robust and targeted anti-tumor response.”

Reviewer #2 comments:

Congratulations for these nice results and very well written manuscript. You demonstrated an approach on how to use an RMF in combination with a gradient field featuring a field free region to locally select the region of interest. You were using MTB as a model microrobot and showed their behavior in vitro and in vivo, for the latter you build a dedicated setup able to accommodate mice.

I have some comments and suggestions, which you can find below as well as some minor corrections within the manuscript.

Response:

We thank the reviewer for the encouraging assessment and insightful comments which helped us improve the quality of our manuscript.

4T/m or 8T/m, respectively are very high gradient strength... I am wondering why you don't observe a magnetic force on the NP and MTBs at the off target regions. Can you comment on this?

Response:

We agree with the reviewer about the implications of having high gradients acting on the MTB and note that the NP tracer particles were not magnetic. As discussed above in comment 1 to reviewer 1, the instantaneous forces supplied by the gradient were much smaller than torque-based or intrinsic propulsion-based forces. Over longer time scales, however, they could still influence the distribution of MTB, and thus also convectively driven co-suspended NPs, in the wells. Indeed, our decision to not proceed further with the 8 T/m configuration was motivated by the potential interference with the RMF-based torque-driven transport. In the case of the 4 T/m configuration, we did still observe subtle gradient effects. However, we performed the experiments under this condition for two primary reasons. Firstly, this condition serves as a stringent test for this actuation scheme. The gradient-based forces primarily exert outward forces at the target point, thereby impeding transport at the target point. In the off-target regions, magnetic forces roughly support the transport in one half and reduce it in the other half (resulting in an average value nearly equivalent to the diffusion-limited scenario i.e., control). Secondly, to ensure RMF uniformity across the five-chambered device, we adhered to the relatively small working volume (approximately 1 cm³) of the magnetic field generator in the miniaturized setup. Although this led to potentially higher gradients from the selection field, it enabled us to eliminate any potential gradient from the electromagnet system. Therefore, we concluded that the observed trade-off (marked by off-target suppression at the cost of reduced on-target transport) can be attributed entirely to the gradients caused by the selection field, which diminish significantly as the setup scales up.

As noted in more detail above in comment 1 by reviewer 1, to further elucidate the role of these gradients, one can compare their order of magnitude with the torque-based and intrinsic propulsion-based forces in living bacteria.

$$F = \nabla(\vec{m} \cdot \vec{B}) \approx 5 \times 10^{-16} \text{ A m}^2 \times 10 \frac{\text{T}}{\text{m}} \approx 5 \times 10^{-15} \text{ N} \equiv 0.005 \text{ pN}$$

The propulsive force for bacteria has been estimated to be 0.1-0.5 pN, with forces arising from the RMF-induced rotation falling in the range of 0.5-1.0 pN (<https://www.science.org/doi/full/10.1126/sciadv.aav4803>). As a result, the contribution of gradient forces is far weaker than the other two sources.

The following section has been added to the discussion to summarize these points:

“The selection field that focuses the transferred torque density is characterized by gradients. In the context of off-target suppression, these gradients help to define spatial resolution, however they can also give rise to forces that

potentially interfere with rotational actuation by drawing bacteria away from a target region. In the limit of weak gradients or microrobots with relatively small, noninteracting magnetic moments that are dominated by torque rather than force, the influence of these pulling forces is small. To assess whether this is the case for the bacteria studied here and shed light on the underlying mechanisms behind the observed effects, it is possible to perform an order-of-magnitude analysis of the forces involved in the studied actuation scheme. The gradient forces in our setups can be estimated in the range of 0.5 to 5 fN. The propulsive forces that drive taxis-driven motion have been reported to be 0.1-0.5 pN.³⁰ And lastly, the forces arising from rotational actuation fall within 0.5-1 pN.³⁰ A comparison between these values implies that the gradient-based forces play a substantially weaker role in the conducted assays relative to the other forces which are key to our actuation scheme. Moreover, the improved spatial selectivity in convection-enhanced NP transport experiments employing the mouse-scale setup reveals diminishing contribution of gradient forces at larger scales.”

It's still not clear to me, why the RMF and selection field leads to a higher concentration of MTB in the tumor cite compared to only RMF. I do agree that due to the combination of these fields, you can choose in which region the MTB are being actuated. Hence, it makes sense to me that you would find more MTB in a tumor inside the field free region than outside, but why are there more MTB in the tumor compared to only using RMF? You also state that these experiments were not significant, but still you argue that you observe an increase. If you don't have statistically significant results, then I would like to see at least a profound explanation.

Response:

We appreciate this critical comment. We showed in our prior studies (<https://www.science.org/doi/abs/10.1126/scirobotics.abo0665>) that torque-based actuation brings about increased extravasation and tissue accumulation. It was possible in that study to inherently limit actuation to the flank tumor employed there using the multi axis electromagnet with a small working volume mentioned above. In the case of a deep-seated tumor, this would not be feasible without also actuating bacteria in surrounding healthy tissue, which triggers extravasation in locations where they are not needed while also reducing the effective dose in circulation that passes the tumor site. This acts as the key motivating factor behind spatially confining the actuation. We have added to the following statement in the revised version to address this limitation:

“While the difference between SF+RMF and control can be attributed to the benefits of torque-driven enhanced transport, higher MTB accumulation compared to global RMF could be explained by differences in net effective dose reaching the tumor site. Global RMF is expected to lead to higher extravasation in off-target areas which limits the available dose for extravasation at the tumor region.”

In terms of our statistical confidence in the observed effects, we found more robust differences in OD measurements meeting the generally accepted definition of statistical significance, while this was not the case for the cfu data. We report both results in the interests of full transparency. Note that both these measurements are meant to assess the number of viable bacteria found in a tumor, however cfu counting is prone to higher variability, which may have obscured the effect. We have added the following to highlight this difference in the text:

“Although the differences in the OD values between conditions do not provide a direct count of the viable MTB population within the tumors, these measurements do reflect initial seeding concentrations and subsequent proliferation. While both cfu counts and OD values assess the number of viable bacteria found in a tumor, the former is prone to higher variability, leading to different levels of statistical confidence.”

I have some concerns about terms used throughout the manuscript, which I would like to address and ask the authors to either change it accordingly or explain why they chose these terms:

1. Magnetic gating field. I did not find where this expression is typically used. To my knowledge these kinds of magnetic fields are called gradient fields showing a field free region/point. You sometimes also name it selection field, which is also a common expression. Why you name it differently is not clear to me. And be consistent, you use all three expressions.

Response:

Thank you for highlighting the usage of different terms for this concept—we do strive for clarity. Thus, we have harmonized the usage of the term “selection field” throughout the manuscript.

2. Focal torque delivery. I am struggling a bit with the expression “delivery” as it is commonly used for e.g. drug delivery. Hence, I would see terms like “application” more appropriate. Also, the expression of “focal point” is strictly/physically seen incorrect or at least misleading since a magnetic field cannot have a focus point.

Response:

We thank the reviewer for these sound suggestions. We have adapted the term and now use ‘focal torque application’. Regarding the expression ‘focal point’, we do believe the usage is justifiable here because we are referring to the spatial distribution of delivered torque density rather than the field itself. The possibility for remote points for of focal torque application is an intriguing and counterintuitive result that is worth emphasizing.

3. Open loop control. Since you even use this term in the title, I was expecting to see some magnetic path guidance within the manuscript. Since, this is not what the article is about I would suggest replacing this expression. My suggestion “Spatially selective delivery of magnetic microrobots” you also don’t show any drug delivery. You could add “to tumor sites” if you like.

Response:

While we believe that the term ‘open loop’ could be justified here because a desired spatial distribution is achieved without active feedback, we agree that for the sake of clarity the term ‘open loop’ might be removed from the title. We have selected a revised title:

Spatially selective delivery of living magnetic microrobots through torque-focusing

Reviewer #3 comments:

In this manuscript, the authors present an actuation scheme based on the combination of a rotating magnetic field typically used for the actuation of artificial bacterial flagella that the authors already used for magnetotactic bacteria, with magnetostatic gating fields. The manuscript is rather technical, the flow is difficult to follow, at least initially, and the authors change the way they present the results between the *in vitro* and the *in vivo*, such that it is uncertain that the *in vivo* treatment is really significantly successful.

Response:

We appreciate the reviewer's critical feedback and careful comments. We endeavoured to address these concerns as fully as possible in an effort to further improve our manuscript.

Regarding the consistency between the presentation of the results *in vitro* and *in vivo*, we hope the clarifications to terminology addressed above in response to reviewer 2 have helped make the manuscript easier to follow and made the connections between experiments more apparent.

Another main point is that the proof of concept of this actuation scheme has already been demonstrated by others in my opinion (see below), and here it rather appears as a follow-up study of the papers the group of authors published recently (DOI: 10.1126/scirobotics.abo0665).

Response:

While we agree that this study is built on torque-based actuation concepts previously explored in our prior publications, here we demonstrate a set of entirely separate findings:

- 1) The conceptual advance of resolution being set by the relative magnitudes of the selection field and RMF
- 2) The influence of torque focusing on the distribution of magnetic microrobots or co-suspended nanoparticles in tissue and tumor models
- 3) The design, construction, and characterization of a unique instrument for focused torque application with a movable field-free point. This has implications on scalability that supports the argument that these methods can be used for deep-seated targets, eventually at human scale.
- 4) An *in vivo* experiment that differs substantially from the paper cited above as it applied an RMF to the entire body of the animal with and without the selection field

Our findings serve to advance magnetically mediated drug delivery strategies for clinical applications. Overcoming the challenge of scalability and translatability has consistently been a bottleneck impeding the progression of magnetic strategies. We think that the new aspects of our findings here hold the potential to lower some of these barriers, aiding in the translation of magnetic delivery techniques for both living and non-living microrobots. In terms of what distinguishes this work from the similar works by others, we have elaborated further in response to the comment about L60-62.

The final main point is the fact that with this method, the authors can effectively drive AMB-1 and enhance particle diffusion locally but by using the torque driven strategy, and specifically by alternating the torque sign, the chemical and aerotactic stimuli are most likely disregarded because the torque driven forces overrule.

Indeed, torque-based actuation overrides intrinsic propulsion, but *only* when torque is being actively applied. Our control strategy is hybrid in the sense that the initial torque-driven enhancement is followed by taxis-driven colonization. Both the *in vitro* spheroid experiments and *in vivo* trial made use of this combination, with 1h of torque-based actuation followed by a window of 24 or 48 h without magnetic stimulus to leverage taxis behavior. Regarding

the alternating torque sign, no form of magnetic field application that we employed alternated the sign of the torque, however, the swept RMF did gradually change the plane of rotation.

While the MTB benefits from having local hypoxic regions around the tumor, this will remain without contribution for the particular technique.

As mentioned above, taxis-based navigation, including aerotaxis to hypoxic niches, is exploited after actuation ceases. Through our magnetic actuation, the initial seeding density of bacteria in and around a tumor is increased, thereby supporting robust colonization. Arguably, this strategy may help to overcome limitations of short-ranged taxis. Bacterial taxis primarily involves the random sampling of the local environment within a confined space. As a result, proximity to the tumor microenvironment is a prerequisite for leveraging this capability to infiltrate deeper regions in response to the local biochemical gradients. The application of magnetic torque effectively reduces barriers to enter the tumor region, facilitating the bacteria's use of local cues to successfully colonize tumors.

The presence of the field free region with relative high field gradients at the borders would indeed most likely always inhibit any aerotaxis or chemotaxis. A discussion around these points would be valuable for future design improvements. Does one need to resign to these taxis to have full control or is it possible to combine them and how compatible is it with the design?

Response:

As an additional detail, the animal or sample is removed from the respective instruments after the 1 hour of magnetic actuation, thus, allowing taxis to take place without the influence of any magnetic field gradient. Moreover, force estimates show that the gradient forces exerted on an MTB in our in vivo setup were considerably lower than intrinsic propulsive forces. Given the average magnetic moment of a single bacterium and the gradients in our in vivo setup (about 1 T/m), the force can be estimated to be:

$$F = \nabla(\vec{m} \cdot \vec{B}) \approx 5 \times 10^{-16} \text{ A m}^2 \times 1 \frac{\text{T}}{\text{m}} \approx 5 \times 10^{-15} \text{ N} \equiv 0.0005 \text{ pN}$$

Assuming a relevant velocity range for MTB, propulsive force can be estimated as 0.1-0.5 pN (<https://www.science.org/doi/abs/10.1126/scirobotics.abo0665>). As a result, the contribution of gradient forces is significantly weaker compared to the other two sources.

We also added the following descriptions in the text to place more emphasis on the hybrid nature of this control scheme:

“This combination can be seen as a hybrid control strategy where magnetically enhanced delivery precedes taxis-driven penetration.”

“Similar to the spheroid experiments, magnetic actuation was followed by taxis-driven motion in this timeframe, reflecting an overall hybrid scheme for tumor colonization.”

Finally, there numerous small flaws that should be modified before publication can be granted. Please find below these more specific points below:

- L14-15, magnetotactic bacteria are described as a biohybrid robotic system. Why not simply a biological one?

Response:

We agree with the reviewer, particularly within the context of this study. We initially used the term "biohybrid microrobots" to reflect their potential as drug carriers if functionalized with cargo. However, for improving clarity in this manuscript, we changed it to "biological robotic system" in the revised version.

"Taking magnetotactic bacteria (MTB) as a model biological microrobotic system for torque-based actuation..."

• L18: *in vitro* is not written in italics. I am uncertain of the guidelines of the journal, but it is not in italics in the text, but in italics in the title and in the captions.

Response:

Thank you for noting the inconsistency in the formatting of these Latin terms. Upon a review of the papers published by the journal, we have ensured that these terms are now presented in a non-italic format in the revised version.

• L47: "deep tissue", should be "deep tissue penetration" or "targeting"?

Response:

Thanks, we have corrected this:

"More recently, uniform fields that steer self-propelling microrobots^{11,20-23} or rotating fields that power motion through applied magnetic torques²⁴⁻²⁹ have offered compelling alternatives that are suited for deep tissue targeting and scalable to patients."

• L60-62: the concept of MPI for not only imaging but also controlling the movement of milli and microsystems is not new. It has already been demonstrated, see for example: <https://doi.org/10.1038/s41598-021-93323-4> or <https://pubs.acs.org/doi/full/10.1021/acsnm.1c00768>

Response:

We thank the reviewer for drawing attention to related studies. In the following, we have presented what sets our work apart from these studies despite of using MPI-inspired actuation principles:

<https://doi.org/10.1038/s41598-021-93323-4>: This study presents a similar scheme for steering a *single* helically shaped microrobot. Such navigation-based methods rely on intermittent feedback regarding the microrobot's position to dynamically adjust actuation parameters in real time. As noted by its authors, this method is sensitive to the influence of blood flow, which can wash away the microrobots. This challenge might be managed in exceptional scenarios such as aneurysm, wherein a balloon catheter can be employed to locally slow blood flow locally. However, in the case of systemic delivery, microrobots will be dispersed quickly throughout the body due to the blood flow. The elegance of our strategy lies in its independence from feedback mechanisms, achieved through the spatially focused application of torque density. This method renders this strategy compatible with numerous *diffuse* systemically administered microrobots, which are most relevant to targeting solid tumors.

<https://pubs.acs.org/doi/full/10.1021/acsnm.1c00768>: This paper introduces a comparable scheme of spatial actuation using selection fields, however, it relies on the formation of aggregates of magnetic nanoparticles, which roll and move within tubular structures. This strategy similarly relies on feedback-dependent steering, which makes it challenging for navigation through complex networks such as the vasculature. Furthermore, swarm effects and aggregation dynamics are substantially counteracted by strong fluid flows and disappear at low local concentrations upon dispersion. Our approach is independent of these effects—from the perspective of a single MTB, our strategy functions regardless of the proximity of its nearest MTB neighbours. This makes our approach well suited for applications *in vivo*.

Notably, neither of these publications nor others that we are aware of have demonstrated spatially selective actuation in vivo using such concepts.

Overall, particularly in the context of drug delivery, combining spatially focused torque density with magnetotactic bacteria concerns applications and possibilities that clearly fall outside the scope of the above approaches. Nevertheless, we do acknowledge the necessity of directing the reader to these two works within the introduction. The following statement has been added to the revised version to address this need:

“More recently, the feasibility of integrating RMF with static selection fields has also been shown.^{31,43–45} In pioneering work, independent control of centimeter-scale screws has been achieved using MPI instrumentation.³¹ Similarly, spatial control of rolling magnetic NP swarms within a tube was investigated.⁴⁴ While these studies demonstrated the compatibility of RMF with selection fields for magnetic actuation, our strategy does not rely on feedback and is compatible with diffuse agents at low concentrations, and thus, well suited for systemic drug delivery.”

Also, we would like to note that we already referenced other works that demonstrated spatially selective torque-based actuation in the following passage:

“To accomplish this, a magnetostatic selection field that supplies a field-free point or field-free line can be introduced, which suppresses torque-based actuation outside these regions.^{30–34}”

• L71: at this stage, the “companion nanoparticles” were not introduced and they are a surprise that is a bit hard to understand.

Response:

We thank the reviewer for pointing us to this. We have addressed this by eliminating the possibly confusing term “companion” in the new version as follows:

“We show how this targeted actuation influences translational velocity of the MTB, convection-enhanced transport of non-magnetic nanoparticles NPs into collagen matrices, and RMF-enhanced colonization of tumor spheroids.”

Further references to “companion NPs” were similarly exchanged with “co-suspended non-magnetic NPs”.

More context

• L81: “a small in vivo study”, what is a “small” in vivo study? What is the interest of a “small in vivo study”?

Response:

The term “small” in this context primarily concerns the size of the cohort used in this study. As this manuscript demonstrates the first prototype for implementing such an actuation scheme with MTB, the in vivo trial here serves as a proof-of-concept study. In order to adhere to 3R principles for animal experimentation, we therefore used small cohorts to test this concept. Notably, an in vivo selection field torque-based actuation experiment has, to the best of our knowledge, never been reported. Nevertheless, we do acknowledge the need to use larger cohorts for follow-up studies with an improved version of the magnetic setup, where tumor treatment will also be examined over an extended timeframe. We have adapted the relevant paragraphs in the discussion to highlight this need:

“Given the promise of this strategy, illustrated by the results of our preliminary in vivo study, follow-up trials with larger cohorts may further elucidate the benefits of targeted torque-driven schemes for both living and synthetic microrobots.”

“In the context of tumor treatment, further insight into aspects other than delivery, such as immune response, is needed before using MTB as a living therapeutic vector. In an optimal scenario, achieving spatially selective delivery holds the potential to better control immune reactions. Furthermore, this effect could be fine-tuned by co-delivering pharmacological agents, thereby orchestrating a robust and targeted anti-tumor response.”

• L165 “green fluorescent NPs”, in the M&M section (1619-620), the NPs are supposed to be red fluorescent. What is right?

Response:

Thank you for pointing this out. Indeed, red fluorescent NPs were used for the experiments (as shown also in the confocal images in Fig. 2F) because of their better dispersion and lower background in green autofluorescent PDMS devices. However, when we initially needed to confirm the proper tissue model compartmentalization in our custom-made chambers (Fig. 2A), we used TAMRA (red) labelled collagen to appropriately visualize the polymerized hydrogel, and thus, had to use another color for the NPs, here green fluorescently labelled ones. We have adapted the relevant sections referring to the choice of fluorescent labels for NPs and collagen used in experiments shown Fig. 2 for improved clarity:

“Successful compartmentalization within the chambers were demonstrated by TAMRA-labeled collagen and a suspension of green fluorescent NPs.”

“For the transport experiments, in contrast to Fig 2A, unlabeled collagen was used in the center of the chambers and MTB suspension containing 200 nm red fluorescent non-magnetic NPs placed in the surrounding compartment, so that fluorescence intensity could be used to infer concentration before and after the actuation with minimal crosstalk due to green background autofluorescence.”

• L200-201 “more than 40 % ... the target region”, why are the error bars so big? Looking at Figure 2G, there is a considerable difference of NP concentration in the target area between +Sweeping RMF -Selection field and +Sweeping RMF +Selection field. Can the author comment on that?

Response:

As indicated within the text, a trade-off exists between on-target transport and off-target suppression due to the gradients opposing transport to the target location. The higher the gradient, the more pronounced the off-target suppression, but at the expense of diminished transport to the targeted area. However, the favourability of this scheme lies in its scalability, which effectively mitigates this trade-off. In an upscaled system, gradients become weaker even in the presence of a relatively strong static field to suppress any off-target actuation (about 1 T/m in our mouse-scale setup compared to 4 or 8 T/m, respectively, in the miniaturized setup used in Fig 2 and 3). As shown in Fig.4 H-J, the mouse scale setup exhibits a more favourable trade-off when compared to the miniaturized configuration.

We note that this topic is better addressed now in the main text as follows:

“One possible source of the reduction in transport at the target point is the role of gradient forces within the experimental setup. Whereas the magnetostatic field applied here is intended primarily to extinguish torque density, its associated gradients can also apply forces to MTB that draw them to the boundary of the chambers and interfere with rotational actuation. The respective average magnitudes of the relevant forces can be estimated to be 0.5 pN for torque based forces³⁰, 0.2 pN for intrinsic bacterial propulsive forces³⁰, and 0.005 pN for magnetic gradient forces (given an upper bound of 10 T/m). Since the gradients in this miniaturized proof-of-concept setup are approximately twice that of the highest gradient realized in the larger setup described in a later section, these forces, while comparatively small, are expected to play a more dominant role here.”

And as noted above in a response to one of the comments from reviewer 1, a more general explanation in the discussion section is added:

“The selection field that focuses the transferred torque density is characterized by gradients. In the context of off-target suppression, these gradients help to define spatial resolution, however they can also give rise to forces that potentially interfere with rotational actuation by drawing bacteria away from a target region. In the limit of weak gradients or microrobots with relatively small, noninteracting magnetic moments that are dominated by torque rather than force, the influence of these pulling forces is small. To assess whether this is the case for the bacteria studied here and shed light on the underlying mechanisms behind the observed effects, it is possible to perform an order-of-magnitude analysis of the forces involved in the studied actuation scheme. The gradient forces in our setups can be estimated in the range of 0.5 to 5 fN. The propulsive forces that drive taxis-driven motion have been reported to be 0.1-0.5 pN.³⁰ Finally, the forces arising from rotational actuation fall within 0.5-1 pN.³⁰ A comparison between these values implies that the gradient-based forces play a substantially weaker role in the conducted assays relative to the other forces which are key to our actuation scheme. Moreover, the improved spatial selectivity in convection-enhanced NP transport experiments employing the mouse-scale setup reveals diminishing contribution of gradient forces at larger scales.”

In the interests of transparency and to adhere to the editorial guidelines of the journal, we have replotted this figure so that individual datapoints are represented. We hope that this also makes the distribution more apparent to readers than only an error bar representing the standard deviation. Please see the response below to the next comment for a reproduced version of the updated Fig. 2G.

• L224-225, figure 2F, it is unclear if the field used is with a constant RMF or a sweeping RMF? The authors should add an example for each type of RMF that they used to make the graph in Figure 2G.

Response:

Thank you for noting this need for additional information in the figure and its caption. We adapted the annotations in the figure and changed the caption to address this:

“(F) Representative images of NP distributions within the collagen core region in target and off-target chambers, under sweeping RMF with and without the application of a selection field.”

Furthermore, we added a new supplementary figure, now fig. S4, to provide examples for the other two conditions, as requested:

Fig. S4: Images of change in NP distribution. The differences in fluorescent intensity between the initial time point and the end point in collagen areas indicate a higher increase under the constant RMF without a selection field (bottom row) in both target and off-target wells, compared to the control (top row)

- L230 “only a non significant reduction in the target chambers is observed”, it may indeed be not statistically significant but still considerable especially given the length of the error bars.

Response:

We agree with the reviewer about this statement. As noted above, a trade-off exists at this scale which becomes less pronounced in the upscaled system, as shown in experiments performed in the mouse-scale setup (Fig.4 H-J). Nevertheless, this trade-off within the same small-scale setup can be better understood by comparing the results of the NP transport with spheroid colonization as they were performed under two different resolutions. To further clarify the correlation between this change in resolution and the shift in the trade-off balance, we have added the following to the main text in the spheroid section:

“To reduce the magnetic gradient forces experienced by the MTB, a consideration explained in the previous section, the device featuring smaller magnets (4 T/m gradient) was again used for these experiments. However, an increased RMF magnitude relative to the magnetostatic field translates to a slightly lower resolution, i.e. a larger field-free region. This change is expected to shift the balance in the trade-off that was observed between off-target suppression and on-target transport.”

“Lastly, it is worth highlighting that the altered resolution within the same setup allowed a similar degree of colonization at the target site to be maintained with or without the selection field. For a miniaturized setup like the one employed here, this comes at the cost of incomplete suppression of off-target effects.”

We note again that in the interests of transparency and to adhere to the editorial guidelines of the journal, the figure panel in question has been replotted with individual datapoints shown so that readers can see the distribution for themselves rather than just an error bar summarizing SD.

- L233 “model of cancer”, a spheroid is not a model of cancer but a model of a solid tumor, as the authors state later on in the text.

Response:

Thanks, we have changed the text accordingly:

“As a physiologically relevant model of solid tumors, the miniaturized setup described in the previous section was adapted to study whether a magnetostatic selection field could spatially control bacterial colonization in tumor spheroids.”

• Fig 3D, can the authors give access to their data as it is difficult to understand how the results +Sweeping RMF - Selection field and +Sweeping RMF +Selection field are significantly different with the error bars depicted in the ++ case.

Response:

Absolutely. The corresponding data file, entitled ‘Dataset 1’, has been submitted as a Supplementary Dataset together with the other documents of our revised version. In addition, we note that the figure panel in question has been replotted to show individual datapoints rather than just error bars. The off-target datapoints are more numerous than the on-target ones because more off-target spheroids were present in the experiment. For convenience, the updated version of this panel is reproduced below:

• L 383, the title is misleading; the authors say the difference is not statistically significant in the text. Therefore, they should not do such a statement in the title of the section (see L421-424).

Response:

As mentioned in the manuscript, we acknowledge the presence of considerable variability in the colony counting data, resulting in statistically nonsignificant differences between the groups. However, the other readout, i.e. OD measurements, showed significant differences between the groups. We believe that given the shape of MTB growth curve in liquid cultures and the relationship between OD_{600} and number density, this method is less susceptible to variability, allowing us to observe differences with greater statistical confidence. Recognizing the absence of consistent confirmation of significance across all readouts, we decided to use the term “improve” to describe the effect without overstating its extent. We hope that the reviewer finds this reasoning justified.

• L387 “cite”, probably a citation is missing here.

Response:

Thank you for noticing this mistake. The relevant citation has been added in the revised version.

“Local application of RMF to subcutaneous flank tumors has previously been shown to improve delivery of MTB into tumors.^{30”}

• L 408-411, In the images the authors show in Figure 5B, MTB seem to appear at deeper areas of the tumour in the global RMF group compared to the SF+RMF group. Maybe more images should be provided to prove the authors’ statement.

Response:

We thank the reviewer for this observation and remark. Below we provide a version of the images the reviewer is referring to with enhanced exposure settings, which might make the broader distribution of MTB across the selected region of the respective tumor slices more apparent for the selection field experiment with SF + RMF vs global RMF. Given the small size of MTB clusters relative to the tumors, we show zoomed-in images of tumor slices to enhance the clarity of bacterial visualization within the tumors. The images in the panel were provided to simply enable visualisation of the MTB within the tumors while analysis for this study was focused primarily on the readout related to the total accumulation in tumors (effective delivered bacterial dose), which we determined via tumor homogenisation and culture of the bacteria therein. Higher bacterial accumulation in tumors treated with SF + RMF are also reflected in these two images, as an overall higher number of MTB/MTB clusters, supported by the intensity analysis plot shown below. However, to refrain from any overstatement regarding penetration depth in the images shown, we have adapted the text as follows:

“Tumors exposed to the combination of selection field and RMF exhibited more clusters of bacteria distributed across the tumor compared to tumors treated with global RMF and control.”

• L 414-416: how many experiments were performed here? No error bars are presented on Fig 5C.

Response:

Individual data points are presented separately in Fig. 5C. We have overall n=4 for the control, n=3 for the global RMF, and n=3 for the SF+RMF. The pooled data are presented in Fig. 5D.

• L442-444: Do the authors have a control with mice without MTB but with an applied magnetic field?

Response:

We did not include a control without MTB but with an applied magnetic field. While we agree about the potential importance of such a control in studies of tumor regression or detailed immune response, we believe that our controls (non-actuated with and without MTB) are most suitable for interpreting the selected readouts within the scope of this study.

- L450-452: if the MTB are really targeting the tumour and not the surroundings, there should be a lower immune response because less MTB are around in the blood flow, can the authors explained their statement again?

Response:

The blood samples were collected 24 hours after administration. According to our research on the immune response towards AMB-1 (<https://www.biorxiv.org/content/10.1101/2023.03.31.535049v1.abstract>) and in line with many other studies investigating the presence of bacteria in the circulation upon systemic injection, it is anticipated that bacteria are cleared from the circulation within the first few hours. This understanding also informed our decision regarding the time window of magnetic actuation, i.e. right within the first hour after systemic administration. Accordingly, the serum cytokine levels can be primarily attributed to inflammation within tissues. The correlation between cytokine levels and bacterial accumulation within tumors, along with more rapid clearance of bacteria from most other organs, suggested to us that higher cytokine levels arise from immune cells encountering the bacteria in the tumor region.

- Figure 5F: the authors should be consistent with the previous graphs and add the * instead of the numerical value. This way, it would be easier for the reader to see, which differences are significant or not.

Response:

Thank you, for consistency we have added stars to the revised version of the figure, reproduced below:

• L553-554: the bacteria are passaged by centrifugation of a culture and putting fresh medium. This is an unconventional way to grow magnetotactic bacteria, for which a 1 to 10% inoculum is typically used. Can the authors explain why they used this peculiar approach to cell culture?

Response:

We acknowledge the need to be more explicit on this point: following resuspension of the pellet, we transfer a fraction of 1 tenth into the new culture, as is common practice. We have added this information in the corresponding M&M section:

“Bacteria were passaged at 1 to 10 ratio every 6-10 days by centrifuging the culture tubes at 9500 rpm for 10 min and resuspending in fresh medium.”

• L619-621: why not using fluorescently-labeled bacteria directly?

Response:

We had two primary motivations for initially employing NPs in our proof-of-concept experiments on a small scale. Firstly, we previously presented torque-driven enhanced convective transport of NPs, and we wanted to study whether this actuation scheme could make this phenomenon spatially selective. And second, since these NPs are non-magnetic, they are not influenced directly by magnetic gradients. This would strengthen the argument that observable difference can be attributed to torque focusing rather than any gradient-based pulling mechanisms. However, as we have noted

above, a trade-off is still present because gradients would still play role by slowly changing the distribution of the torque-actuators, i.e. MTB, within the device. We have adapted the text to clarify our purpose:

“Because we have previously shown that convection generated by MTB subjected to RMFs can observably increase the transport of co-suspended NPs into collagen matrices, we hypothesized that this effect could be spatially restricted through the superposition of a selection field. Additionally, since the NPs were nonmagnetic, their enhanced transport are predominantly governed by torque-induced fluid convection, highlighting the impact of torque focusing compared to any gradient-based pulling.”

REVIEWERS' COMMENTS

Reviewer #2 (Remarks to the Author):

Dear authors,

congratulations again for this impressing study and nice results. For me, all comments have been addressed accordingly and I recommend the manuscript for publication.

Two typos I noticed: Don't forget to also change the title in the SI file and Figure S10 there is an incomplete sentence in the caption.

I also carefully read through the comments of reviewer #1. From my point of view the answers from the authors are sound and the manuscript has been edited accordingly.

Reviewer #3 (Remarks to the Author):

In the revised version of the manuscript, Mirkhani, Schuerle et al. proposed additional clarification of the work they performed. However, they fail to convinced me that the technique they propose is a major step towards translation as compared to their recent publication mentioned in the first review round. This is in particular true because their in vivo results are not convincing as pointed out by all referees. The new version of the manuscript is not bringing any new light on that aspect, also because the authors have decided not to change the title of their paragraph (l 416).

In addition, the referees were struggling with numerous terminology employed in the manuscript, and "delivery" was one of these, and again, the authors have decided to keep this term, and even placed it in the title of the paper. Another simple example is "translational velocity of MTB": can the authors not simply write that the bacteria "roll"? I think the authors should write that here, what they do is that they have a rotating field that turns around the tumor and thereby brings the bacteria in closed vicinity. Once this is done, they let nature (taxis) do the rest. This would be understandable, much better than "torque-driven surface exploration".

There are many little things remaining, some of them reported below:

- L 76: what are the possible sizes?

- L 91: i insist, “small” should be removed, either the authors are convinced by their work and should present it or they are not and should not, but adding “small” is not scientifically relevant and will not help.
- L 263: it is nice to say 25 μ L of bacteria, but a concentration would be more relevant and usefull for reproducibility.
- L 485 and further: as pointed out earlier, if the selection field worked effectively, there should be less bacteria outside of tumors and a lower immune response. I do not see any change by the authors, and I do not understand the advantage of using the selective field if the systemic immune response is higher. Apparently, the authors share this opinion if I understand correctly what is written on L 597.
- The title of the paper for the SI (I 947) is different from the title of the paper at the beginning (I 1).

Response Document for *NCOMMS-23-29093B*

“Spatially selective delivery of living magnetic microrobots through torque-focusing”

We thank the reviewers and editorial team for their helpful comments and requests. All new changes have been highlighted in turquoise and previous changes remained marked in yellow. We have also provided a clean article version without tracked changes and highlights.

REVIEWERS' COMMENTS

Reviewer #2 (Remarks to the Author):

Dear authors,

congratulations again for this impressive study and nice results. For me, all comments have been addressed accordingly and I recommend the manuscript for publication.

We thank the reviewer for this positive feedback and appreciate his or her comments, which have helped us to improve our manuscript.

Two typos I noticed: Don't forget to also change the title in the SI file and Figure S10 there is an incomplete sentence in the caption.

Thanks for bringing these to our attention. The title in the SI has been updated and the mentioned caption has been completed as follows:

“Assembled magic sphere for magnetic characterization is shown. A cube of Hall sensors located at the center of the sphere measures the selection field from the magnets in a 3D volume.”

I also carefully read through the comments of reviewer #1. From my point of view the answers from the authors are sound and the manuscript has been edited accordingly.

We thank the reviewer for his or her assessment and the time he or she spent in additionally considering our responses to Reviewer 1.

Reviewer #3 (Remarks to the Author):

In the revised version of the manuscript, Mirkhani, Schuerle et al. proposed additional clarification of the work they performed. However, they fail to convince me that the technique they propose is a major step towards translation as compared to their recent publication mentioned in the first review round. This is in particular true because their *in vivo* results are not convincing as pointed out by all referees. The new version of the manuscript is not bringing any new light on that aspect, also because the authors have decided not to change the title of their paragraph (l 416).

We thank the reviewer for his or her comments and candour. It is regrettable that the changes we implemented have not fully convinced him or her. With regard to the *in vivo* results, as mentioned before, given the formal statistical significance in only one readout (OD₆₀₀ of liquid cultures), we tried

to tone down claims and simply report the results in a transparent and comprehensive manner. We have now also adapted the title of the paragraph in question accordingly:

“Torque-focusing shows promise for improved bacterial tumor colonization in vivo”

Regarding the value of smaller in vivo experiments, firstly, pilot experiments that measure multiple experimental readouts and reveal consistent trends arguably do offer scientific value, albeit preliminary, provided that it is reported clearly and transparently, as we have done. To the best of our knowledge, this is the first in vivo delivery experiment with RMFs and selection fields that has been attempted. Beyond the immediate scientific value, pilot experiments also inform the design of subsequent larger in vivo experiments and in our case are needed to help convince the regulatory bodies that it is worthwhile to perform these experiments with higher statistical power in the future.

In addition, the referees were struggling with numerous terminology employed in the manuscript, and “delivery” was one of these, and again, the authors have decided to keep this term, and even placed it in the title of the paper.

We acknowledged the potentially confusing terminology in the first round of revisions and attempted to make changes to improve clarity. In particular, the term “delivery” was suggested to be confusing specifically *when applied to torque density*. In reviewing the manuscript, we have found instances where this had not been replaced and thank the reviewer for bringing this to our attention. As a result, we consistently now use the terms “application” and “transferring” in reference to torque. On the other hand, “delivery” is now consistently used in reference to MTB/agents/microrobots/drugs. Considering that the concept of microrobots and nanocarriers frequently occurs in the literature in conjunction with “drug delivery”, in the sense of getting compounds, materials, or agents to a particular targeted site, we would argue that this usage is clear and comprehensible. Please note “delivery” in the revised title is used in that sense.

Another simple example is “translational velocity of MTB”: can the authors not simply write that the bacteria “roll”?

One of our previous publications (<https://onlinelibrary.wiley.com/doi/full/10.1002/adfm.202003912>) demonstrated that suspensions of MTB exposed to RMF can exhibit volumetric flows. This means that bacteria located away from the surface (thus clearly *not* rolling on it) can also exhibit translational motion, and their velocity does not necessarily match the rolling velocity due to the interactions between neighbouring rotating bacteria. As a result, we find “translational velocity of MTB” to be a more accurate term in this context. We appreciate the reviewer’s attention to this important detail and acknowledge that this suggestion would have improved the manuscript if the MTB were indeed simply rolling—our goal has been to state things as simply as possible while remaining accurate.

I think the authors should write that here, what they do is that they have a rotating field that turns around the tumor and thereby brings the bacteria in closed vicinity. Once this is done, they let nature (taxis) do the rest. This would be understandable, much better than “torque-driven surface exploration”.

We thank the reviewer for this comment, and for encouraging us to think carefully about opportunities to improve concision and clarity. Nevertheless, we believe that the phrasing suggested above does not reflect our understanding of the process it describes. We agree that initial magnetically enhanced accumulation is followed by taxis-driven motion and have explicitly stated this repeatedly in the text. However, it is worth highlighting again that, due to the nature of rotating fields and small field gradients in the selection field, there is no appreciable *force* that would pull (or “bring”) the bacteria toward the tumor site. Rather, the rotational character of the field, which has been shown to enhance extravasation and tissue penetration, is restricted (or focused) to the tumor site by the selection field. The main

proposed mechanism behind enhanced extravasation and penetration under rotational fields is torque-driven surface exploration. (<https://www.science.org/doi/full/10.1126/scirobotics.abo0665>).

There are many little things remaining, some of them reported below:

- L 76: what are the possible sizes?

As demonstrated by our numerical results, the resolution of the torque focusing effect depends on the relative magnitude of the RMF and selection field. This implies that if a selection field with a strong gradient is used and if the required RMF can be made arbitrarily small, then the focal point could also be made arbitrarily small. Conversely, a selection field with a weak gradient and a very strong RMF would essentially provide no spatial selection. We recently experimentally corroborated this relationship using low frequency inductive sensing to detect torque transfer to a model permanent micromagnet in glycerol. See Fig 4G of <https://doi.org/10.1093/pnasnexus/pgad297> The generality of this relation is why it is most precise and accurate to state the resolution in terms of the relative magnitude of the selection field and the RMF. In practice, in the present manuscript, our experiments at two scales (miniaturized setup and mouse-scale apparatus) show that given the range of RMF magnitudes used for MTB (10-20 mT), mm- to cm-scale resolution can be readily achieved. We have slightly modified the text to better convey this feature:

“In this method, the spatial focusing of torque density reduces the problem of targeted actuation to adjusting the size and position of the field-free point with respect to the targeted tissue, eliminating the need to individually track microrobots.”

- L 91: i insist, “small” should be removed, either the authors are convinced by their work and should present it or they are not and should not, but adding “small” is not scientifically relevant and will not help.

We acknowledge the reviewer’s concern on this matter. Nonetheless, we believe that communicating the limitations of our study clearly is indeed scientifically relevant. Our in vivo study, which we reiterate is apparently the first of its kind attempted, shows promising results for using this spatially selective strategy. However, we are fully aware that additional statistical power is desirable in the future. As pointed out in the discussion section, repeating the experiments in immune-competent animals with larger cohorts and for longer time periods could shed more light on the extent of advantages provided by our technique.

- L 263: it is nice to say 25 μ L of bacteria, but a concentration would be more relevant and usefull for reproducibility.

We agree with the reviewer that concentration is important in this context—together the concentration and volume indicate the dose. The concentration was indeed mentioned in the Methods section under “MTB culture and fluorescent labelling”. However, after revisiting the subsections in the Methods section, we acknowledge that this parameter was harder to find than it ought to be. Therefore, we moved the corresponding part to the “In vitro spheroid colonization under torque-focusing”:

“To stain the bacteria and allow tracking of subsequent generations, 2 μ l of a far-red proliferative dye (CellTrace™ Far Red Cell Proliferation Kit, ThermoFisher) was added to 1 ml of bacteria suspension at 5×10^8 cells/ml, assessed by optical density measurements. Following 20 min of agitation on a shaker while protected from light, the dye was deactivated using 100 μ l of DMEM for 10 min. The bacteria were spun down and resuspended in 1 ml DMEM for subsequent experiments.”

As a result of this, we slightly adapted the titles in the Methods section to facilitate finding relevant parameters for the readers. We thank the reviewer for taking time to carefully read our methods sections and help us find ways to further improve it.

- L 485 and further: as pointed out earlier, if the selection field worked effectively, there should be less bacteria outside of tumors and a lower immune response. I do not see any change by the authors, and I do not understand the advantage of using the selective field if the systemic immune response is higher. Apparently, the authors share this opinion if I understand correctly what is written on L 597.

In L485 and further of the results section, we report the cytokine levels found in experimental and control groups. As briefly discussed in the previous response document, the conclusion we can draw from the cytokine levels depends on specifically *when* they are measured. The assumption of the reviewer would make most sense for earlier time points. In another study from our group (<https://www.biorxiv.org/content/10.1101/2023.03.31.535049v1.abstract>), we observed significant clearance of bacteria from blood within 3 hours in vitro. Although the dynamics could be different in vivo, this finding along with the reported MTB biodistribution in literature suggests that bacteria outside of the tumor might less heavily contribute to the cytokine level after 24 hours. Similar trends between the cytokine levels and the amounts of intratumoral bacteria is also possibly in line with this conclusion.

However, it is worth noting that the main goal of this measurement was to assess the general safety of the method and while we see higher levels for the selection field group, they do not reflect a dramatic increase compared to controls. We acknowledge that more and longer time points would be needed for obtaining a more comprehensive picture of the situation and that it would make sense to incorporate this into future studies.

In L597 of our discussion we argue: *“In an optimal scenario, achieving spatially selective delivery holds the potential to better control immune reactions.”* We emphasize that this is a discussion point where we reflect on ideal scenarios that could include reduction of dosages due to improved agent delivery, and thus, overall reduced global immune stimulation (through bacteria in circulation) and a desirable more local immune stimulation in the tumor.

- The title of the paper for the SI (l 947) is different from the title of the paper at the beginning (l 1).

Thank you for noticing this. It is revised now.